# A non-bactericidal cathelicidin provides prophylactic efficacy against bacterial infection by driving phagocyte influx

Yang Yang[1†], Jing Wu[2†], Qiao Li[1], Jing Wang[1], Lixian Mu[2], Li Hui[3], Min Li[1], Wei Xu[1]*, Hailong Yang[2]*, Lin Wei[1]*

[1]Jiangsu Provincial Key Laboratory of Infection and Immunity, Institutes of Biology and Medical Sciences, Soochow University, Suzhou, China; [2]School of Basic Medical Sciences, Kunming Medical University, Kunming, China; [3]The Affiliated Guangji Hospital of Soochow University, Suzhou, China

**Abstract** The roles of bactericidal cathelicidins against bacterial infection have been extensively studied. However, the antibacterial property and mechanism of action of non-bactericidal cathelicidins are rarely known. Herein, a novel naturally occurring cathelicidin (*Popu*CATH) from tree frog (*Polypedates puerensis*) did not show any direct anti-bacterial activity in vitro. Intriguingly, intraperitoneal injection of *Popu*CATH before bacterial inoculation significantly reduced the bacterial load in tree frogs and mice, and reduced the inflammatory response induced by bacterial inoculation in mice. *Popu*CATH pretreatment also increased the survival rates of septic mice induced by a lethal dose of bacterial inoculation or cecal ligation and puncture (CLP). Intraperitoneal injection of *Popu*CATH significantly drove the leukocyte influx in both frogs and mice. In mice, *Popu*CATH rapidly drove neutrophil, monocyte/macrophage influx in mouse abdominal cavity and peripheral blood with a negligible impact on T and B lymphocytes, and neutrophils, monocytes/macrophages, but not T and B lymphocytes, were required for the preventive efficacy of *Popu*CATH. *Popu*CATH did not directly act as chemoattractant for phagocytes, but *Popu*CATH obviously drove phagocyte migration when it was cultured with macrophages. *Popu*CATH significantly elicited chemokine/cytokine production in macrophages through activating p38/ERK mitogen-activated protein kinases (MAPKs) and NF-κB p65. *Popu*CATH markedly enhanced neutrophil phagocytosis via promoting the release of neutrophil extracellular traps (NETs). Additionally, *Popu*CATH showed low side effects both in vitro and in vivo. Collectively, *Popu*CATH acts as a host-based immune defense regulator that provides prophylactic efficacy against bacterial infection without direct antimicrobial effects. Our findings reveal a non-bactericidal cathelicidin which possesses unique anti-bacterial action, and highlight the potential of *Popu*CATH to prevent bacterial infection.

*For correspondence:
xuweifd828@126.com (WX);
jxauyhl@163.com (HY);
weilin1005@126.com (LW)

†These authors contributed equally to this work

Competing interest: The authors declare that no competing interests exist.

## Editor's evaluation

This manuscript describes for the first time a novel cathelicidin, namely, *Popu*CATH which is able to prevent the development of infection by different bacterial species in two different animal models, frog and mouse. The mechanism of action is exerted through priming of neutrophil efflux.

## Introduction

Antimicrobial peptides (AMPs) are a wide array of gene-encoded small defensive molecules that have been identified from prokaryotic to eukaryotic kingdoms, including bacteria, fungi, plantae, and animalia (*Mygind et al., 2005*; *Radek and Gallo, 2007*; *Silva et al., 2014*; *Zhang and Gallo,*

2016). In vertebrates, cathelicidins constitute one of the major antimicrobial peptide families (*Wei et al., 2013*). Cathelicidins are composed of an N-terminal signal peptide (about 30 amino acids), a highly conserved cathelin domain (99–114 residues) between signal peptide and mature peptide, and a C-terminal mature peptide (12–100 residues) with diverse structures (sequence and length) and functions (*Zanetti et al., 2000*). Based on the structural characterisation, the mature peptides of cathelicidins can be distinguished into amphipathic $\alpha$-helical structure (e.g. human LL37), beta-sheet structure (e.g. porcine protegrin), and structure enriched in specific amino acids like proline/arginine residues (e.g. bovine Bac5 and Bac7; *Zanetti et al., 2000*). Cathelicidins were initially characterised for their direct antimicrobial activity (*Gennaro et al., 1989*), which act as natural amino acid-based antibiotics with broad spectrum that directly target bacteria (*Snoussi al., 2018*). Due to their rapid and potent bactericidal property without significant toxicity, cathelicidins have been considered as promising peptide antibiotics for therapy of bacterial infection (*Oyston et al., 2009*; *Zanetti, 2004*). Several cathelicidin-derived peptide antibiotics have been tested in clinical trials (*Gordon et al., 2005*; *Mwangi et al., 2019*). In addition to direct antimicrobial activity, more and more studies demonstrated that cathelicidins possess diverse immunomodulatory activities (*Sun et al., 2015a*; *Zhang et al., 2015*).

Since the first purification of cathelicidins (Bac5 and Bac7) from bovine neutrophils (*Gennaro et al., 1989*), more than 1500 vertebrate cathelicidins have been identified from aquatic vertebrates to terrestrial vertebrates, including fishes, reptiles, amphibians, birds, and mammals (https://www.ncbi.nlm.nih.gov/protein/?term=cathelicidin). Amphibians, the evolutionary link of vertebrates from aquatic animals to terrestrial animals, possess an ancient but powerful innate immune system to thrive in a wide range of habitats (*Xu and Lai, 2015*). Gene-encoded AMPs form a first line of innate immunity in amphibians to defense noxious microbes (*Li et al., 2007*). In the last decades, a total of 8350 (Jun 21, 2021, https://amphibiaweb.org/) amphibian species have been documented, and more than 1900 AMPs have been identified from amphibians (*Xu and Lai, 2015*). However, cahtelicidins were absent in amphibians until cathelicidin-AL was characterised from *Amolops loloensis* (Anura: Ranidae), which filled the evolutionary gap of cathelicidin in vertebrates (*Hao et al., 2012*). So far, about 20 cathelicidins were identified from amphibians, including frog cathelicidins identified from *Amolops loloensis* (*Hao et al., 2012*), *Paa yunnanensis* (*Wei et al., 2013*), *Rana catesbeiana* (*Ling et al., 2014*), *Limnonectes fragilis* (Anura: Ranidae) (*Yu et al., 2013*), *Microhyla heymonsivogt* (Anura: Microhylidae) (*Chai et al., 2021*), *Polypedates puerensis* (Anura: Rhacophorinae) (*Mu et al., 2017*), toad cathelicidins identified from *Duttaphrynus melanostictus* (*Gao et al., 2016*), *Bufo bufo gargarizans* (Anura: Bufonidae) (*Sun et al., 2015b*), salamander cathelicidin identified from *Tylototriton verrucosus* (*Mu et al., 2014*), *Andrias davidianus* (*Yang et al., 2017*) (Caudata: Salamandridae), and others. Most of these cathelicidins from frogs, toads, and salamanders exhibited direct antimicrobial activities with broad spectrum via dual bactericidal-immunomodulatory activities. For example, cathelicidin-PY and cathelicidin-PP showed bactericidal activity and anti-inflammatory activity by disrupting bacterial membrane and blocking TLR4-mediated inflammatory response, respectively (*Mu et al., 2017*; *Wei et al., 2013*).

Overall, the anti-infective action and relative mechanism of bactericidal cathelicidins have been extensively studied. However, the role and mechanism of action of non-bactericidal cathelicidins against bacterial infection remain unknown. In this study, a novel naturally occurring glycine-rich cathelicidin, designated as *Popu*CATH, was identified from the tree frog of *P. puerensis*. *Popu*CATH did not show any direct antimicrobial activities. Interestingly, intraperitoneal injection of *Popu*CATH effectively prevented bacterial infection in tree frogs and mice, indicating an indirect antimicrobial mechanism of *Popu*CATH. The mechanism of action was investigated both in vitro and in vivo. Our study provides new insight for better understanding the anti-infective property and relative mechanism of non-bactericidal cathelicidin, and highlights a host-based immune defense regulator for preventing bacterial infection without drug-resistant risk.

## Results

### A novel naturally occurring cathelicidin was identified from the skin of tree frog, *P. puerensis*

To understand the peptidomics of *P. puerensis* skin, the skin secretions were firstly separated by molecular sieving fast pressure liquid chromatography (FPLC) as indicated in *Figure 1—figure supplement 1A*. The eluted peak containing the objective peptide in this study (marked by an arrow) was further purified by a reversed-phase high-performance liquid chromatography (RP-HPLC) C18 column for two times (*Figure 1—figure supplement 1B and C*, marked by an arrow). The purified peptide exhibited an observed molecular weight of 4295.9 Da (*Figure 1—figure supplement 1D*). Then, a total of 16 amino acids at N-terminus were determined as SRGGRGGRGGGGSRGG by automated Edman degradation. The N-terminus is enriched in glycine residues, which is possibly a novel member of cathelicidin antimicrobial peptides like those glycine-rich cathelicidins found in frog (*Hao et al., 2012*) and fish (*Broekman et al., 2011*).

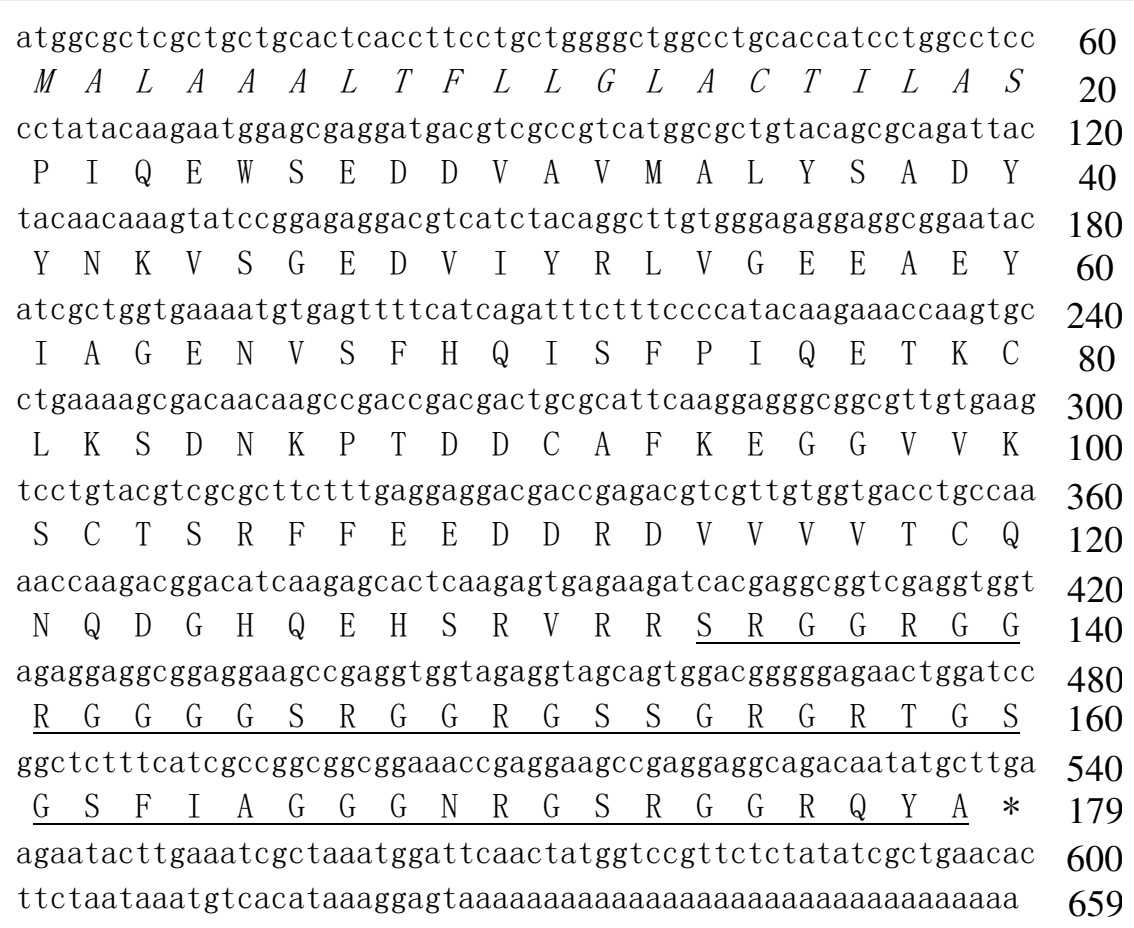

**Figure 1.** The nucleotide sequence encoding the precursor of *Popu*CATH and the deduced amino acid sequence. (**A**) The amino acid sequence of mature peptide is underlined, and the putative signal peptide is italic. Asterisk (*) indicates stop codon. Deduced amino acid sequence of *Popu*CATH precursor was translated in ExPASy Translate Tool (http://web.expasy.org/translate/). Sequence Blast was performed with Blastx (https://blast.ncbi.nlm.nih.gov/Blast.cgi).

The online version of this article includes the following figure supplement(s) for figure 1:

**Figure supplement 1.** Peptide purification.

**Figure supplement 2.** Multiple alignment of the precursor of *Popu*CATH with other cathelicidins.

**Figure supplement 3.** Phylogenetic analysis of the precursor of *Popu*CATH with other vertebrate cathelicidins.

**Figure supplement 4.** Secondary structural analysis.

According to this implication, we designed primer based on the conserved region of amphibian cathelicidins to clone the gene encoding the objective peptide. The nucleotide sequence (GenBank accession number: KY391886) encoding the precursor of the objective peptide was cloned from the skin cDNA library (*Figure 1*). The coding sequence of the precursor included 659 nucleotides that encodes a precursor containing 179 amino acid residues (*Figure 1*). The full-length amino acid sequence of the mature peptide (designated as *Popu*CATH) was determined as shown in *Figure 1*. BLAST comparison confirmed that the precursor of *Popu*CATH is definitely a novel member of cathelicidin antimicrobial peptide family, which shares a highly conserved signal peptide and cathelin domain at N-terminus with amphibian cathelicidins (*Figure 1—figure supplement 2A*). Phylogenetic tree analysis indicated that *Popu*CATH combined with amphibian and fish cathelicidins form the second cluster, showing close evolutionary relationship with amphibian cathelicidins and fish cathelicidins (*Figure 1—figure supplement 3*).

Primary structural analysis indicated that *Popu*CATH is composed of 46 amino acid residues, including 41 polar residues and 5 non-polar residues, which is a glycine-rich cathelicidin (21 glycine residues) like those found in frog and fish (*Figure 1—figure supplement 2B*, *Supplementary file 1*). *Popu*CATH has net charges of +10 and a theoretical isoelectric point of 12.60 (*Supplementary file 1*). The theoretical molecular weight is well matched with the observed molecular weight (*Figure 1—figure supplement 1*, *Supplementary file 1*). Secondary structural analysis indicated that *Popu*CATH mainly adopts random coil confirmation in both aqueous solution and membrane-mimetic solution (*Figure 1—figure supplement 4*, *Supplementary file 2*).

## *Popu*CATH lacks direct antimicrobial activity but can prevent bacterial infection in tree frogs

Cathelicidins were initially described for their direct antimicrobial activity (*Gennaro et al., 1989*). Therefore, we first detected the direct antimicrobial activity of *Popu*CATH in vitro by MIC assay. To our surprise, *Popu*CATH didn't show any antimicrobial activity against the tested bacteria (a total of 40 strains) at the concentration up to 200 µg/mL, including Gram-negative bacteria, Gram-positive bacteria, fungi, and aquatic pathogenic bacteria (*Supplementary file 3*). Similarly, in time-kill assays, 200 µg/mL of *Popu*CATH did not reduce the CFUs of *E. coli*, *S. aureus*, *C. albicans*, and *A. hydrophila* after incubation for 1, 2, 3, and 4 hr, respectively (*Figure 2A*). Furthermore, 200 µg/mL of *Popu*CATH did not alter bacterial metabolic activity during the exponential growth phase of *E. coli*, *S. aureus*, *C. albicans*, and *A. hydrophila* after incubation for 1, 2, 3 and 4 hr, respectively (*Figure 2B*). Cathelicidins are usually membrane-active agents which can alter the surface morphology of bacteria (*Wei et al., 2013*). As shown in *Figure 2C*, 200 µg/mL of *Popu*CATH did not alter the surface morphology of *E. coli* and *S. aureus* after *Popu*CATH treatment. While the positive control peptide PY (1× MIC, cathelicidin-PY), a previously described amphibian cathelicidin from *P. yunnanensis* (*Wei et al., 2013*) markedly inhibited bacterial growth (*Supplementary file 3*), showed bactericidal activity (*Figure 2A*), reduced bacterial metabolic activity (*Figure 2B*), and altered bacterial surface morphology (*Figure 2C*). These results indicated that *Popu*CATH lacks direct antimicrobial activity.

In order to understand whether *Popu*CATH has antimicrobial activity in vivo, *Popu*CATH (10 mg/kg) was intraperitoneally injected into *P. puerensis* 8 hr, or 4 hr prior to (–8 hr or –4 hr), or 4 hr after (+ 4 hr) intraperitoneal bacterial inoculation, and the bacterial load was recorded. Compared to PBS treatment, *Popu*CATH (10 mg/kg) treatment at 8 hr or 4 hr before bacterial inoculation significantly reduced the bacterial load in tree frogs, but *Popu*CATH (10 mg/kg) treatment at 4 hr after bacterial inoculation did not significantly reduce the bacterial load (*Figure 2D*), indicating that pretreatment with *Popu*CATH significantly prevented bacterial infection in tree frogs.

## *Popu*CATH exhibits low toxic side effects to mammalian cells and mice

In order to further investigate the mechanism of action of *Popu*CATH against bacterial infection in vivo, it was necessary to move from a frog system to a mouse system. Thus, the toxicity of *Popu*CATH to mammalian cells and mice were evaluated. At concentration up to 200 µg/mL, *Popu*CATH didn't show any cytotoxicity to mouse peritoneal macrophages and humane monocyte THP-1 cells (*Figure 3A*), and didn't show any hemolytic activity to mouse erythrocytes and rabbit erythrocytes (*Figure 3B*).

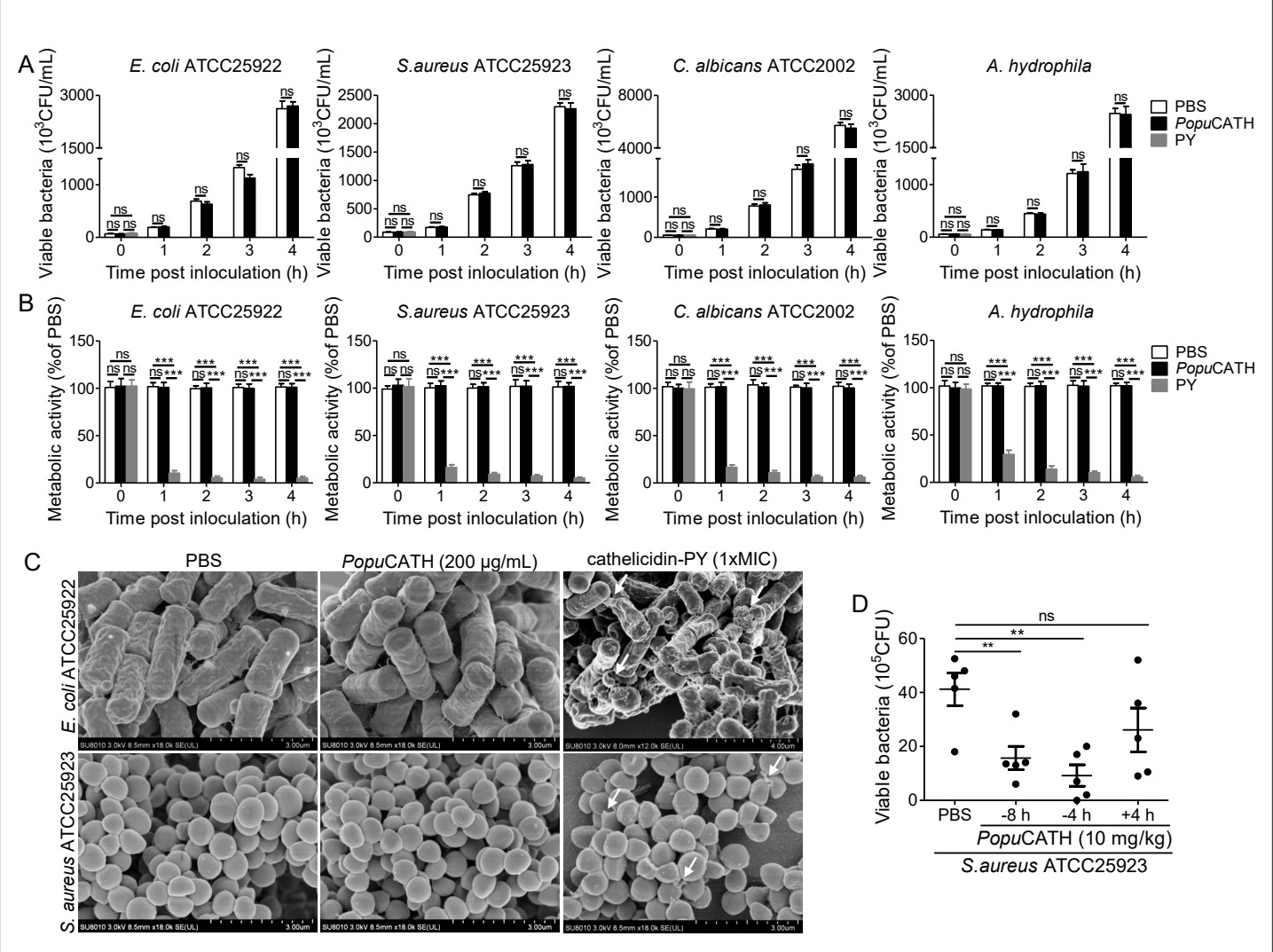

**Figure 2.** *PopuCATH* lacks direct antimicrobial activity but can prevent bacterial infection in tree frogs. (**A**) Bacterial killing kinetic assay. *E. coli*, *S. aureus*, and *C. albicans* were diluted in Mueller-Hinton broth, and *A. hydrophila* were diluted in nutrient broth at density of $10^5$ CFU/mL. *PopuCATH* (200 μg/mL), PY (cathelicidin-PY, 1× MIC, positive control) or PBS was added and incubated at 37℃ or 25℃. At indicated time points, the CFUs were counted. (**B**) Microbial metabolic activity assay. *E. coli*, *S. aureus*, *C. albicans*, and *A. hydrophila* were diluted in DMEM at density of $10^5$ CFU/mL, and *PopuCATH* (200 μg/mL), PY (cathelicidin-PY, 1× MIC, positive control) or PBS was added. Microbial dilution (100 μL/well) and WST-8 (10 μL/well) was added to 96-well plates, respectively, and incubated at 37℃ or 25℃. At indicated time points, absorbance at 255 nm was monitored. Metabolic activity was expressed as the percentage of the PBS-treated group. (**C**) SEM assay. *E. coli* ATCC25922 and *S. aureus* ATCC25923 were washed and diluted in PBS ($10^5$ CFU/mL). *PopuCATH* (200 μg/mL), PY (1× MIC, positive control) or PBS was added into the bacterial dilution and incubated at 37℃. After incubation for 30 min, bacteria were centrifuged at 1000 *g* for 10 min, and fixed for SEM assay. The bacterial surface morphology was observed using a Hitachi SU8010 SEM. (**D**) Anti-bacterial activity in tree frogs. *PopuCATH* (10 mg/kg) was intraperitoneally injected into *P. puerensis* (n = 5, 21–30 g) at 8 or 4 hr prior to (–8 or –4 hr), or 4 hr after ( + 4 hr) *S. aureus* ATCC25923 inoculation ($10^8$ CFU/frog, intraperitoneal injection). At 18 hr post bacterial challenge, peritoneal lavage was collected for bacterial load assay. **p < 0.01, ***p < 0.001, ns, not significant.

The immunogenicity of *PopuCATH* was evaluated by determination of its proliferative capacity on mouse lymphocytes isolated from mesenteric lymph node (MLN) and spleen. As shown in *Figure 3C*, *PopuCATH* did not induce proliferation of lymphocytes isolated from mouse MLN and spleen at any dose tested up to 200 μg/mL, unlike the positive control ConA.

Hypersensitivity to *PopuCATH* was assessed in a mast cell degranulation assay. Mast cells modulate immediate hypersensitivity reactions and nonspecific inflammatory reactions (*Scott et al., 2007*). *PopuCATH* did not induce mast cell degranulation at any dose tested up to 200 mg/mL (*Figure 3D*),

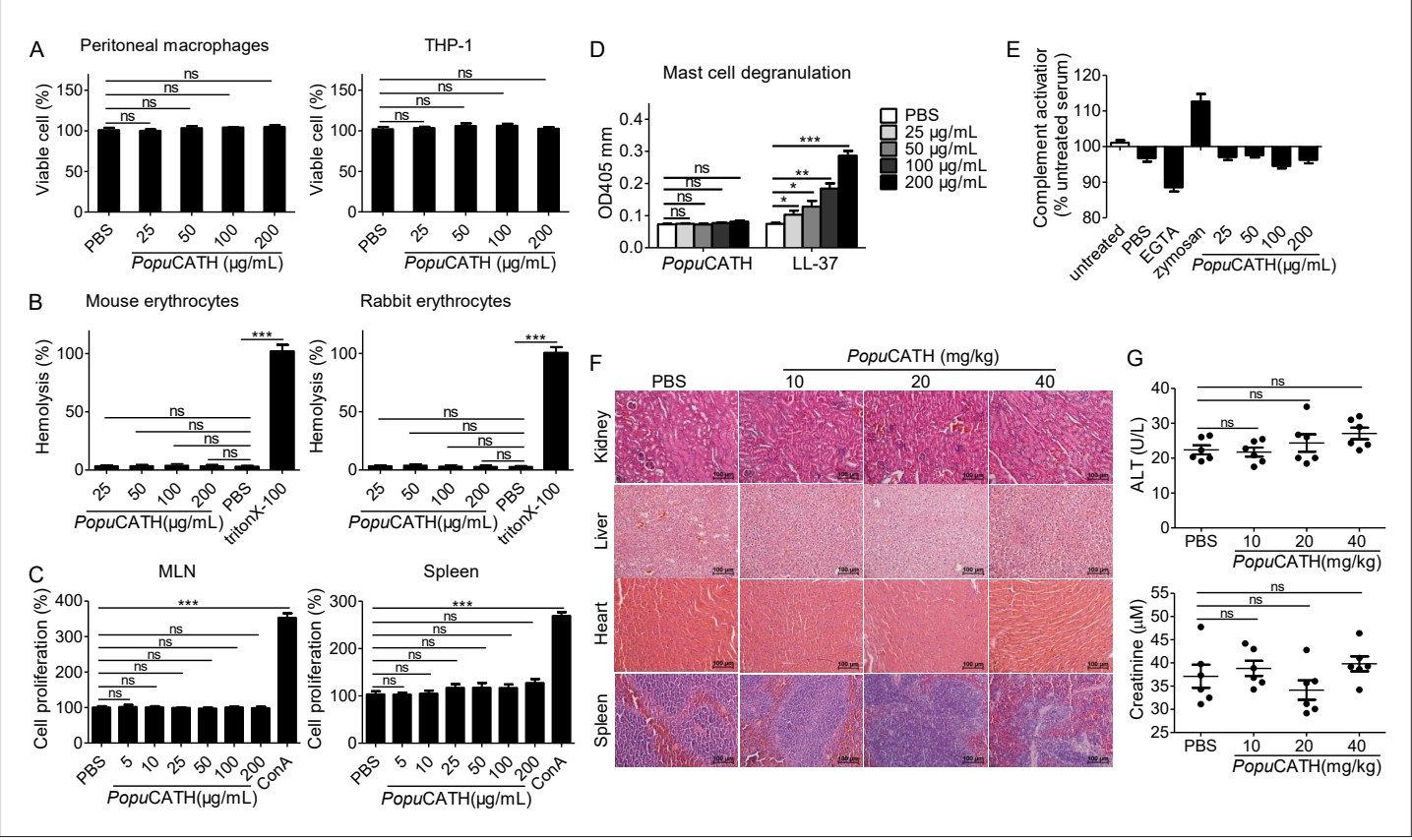

**Figure 3.** *Popu*CATH exhibits low toxic side effects to mammalian cells and mice. (**A**) Cytotoxicity assay. Peritoneal macrophages or THP-1 cells were seeded in 96-well plates (5 × 10⁵ cells/well, 200 µL). *Popu*CATH (25, 50, 100, or 200 µg/mL) was added and cultured for 24 hr. CCK-8 reagent (10 µL/ well) was added and incubated for 1 hr. The absorbance at 450 nm was recorded. Viable cells was expressed as the percentage of the PBS-treated group. (**B**) Hemolysis assay. Mouse erythrocytes and rabbit erythrocytes were washed with 0.9% saline and incubated with *Popu*CATH (25, 50, 100, and 200 µg/mL), PBS or triton X-100 (1%, positive control) at 37°C. After incubation for 30 min, the erythrocytes were centrifuged at 1000 *g* for 5 min. The absorbance at 540 nm was measured. Hemolytic activity was expressed as the percentage of the triton X-100-treated group. (**C**) Immunogenicity assay. Lymphocytes isolated from mesenteric lymph node (MLN) and spleen of mice were suspended in RPMI 1640 (2%FBS) and seeded in 96-well plates (5 × 10⁴ cells/well, 200 µL). *Popu*CATH (25, 50, 100, or 200 µg/mL) or concanavalin A (Con A, 2 µg/mL) was added and incubated at 37°C for 24 hr. CCK-8 reagent (10 µL/well) was added. After incubation at 37°C for 1 hr, the absorbance at 450 nm was measured. Cell proliferation was expressed as the percentage of PBS-treated group. (**D**) Hypersensitivity assay. RBL-2H3 cells were seeded in 96-well plates (2 × 10⁴ cells/well, 200 µL) and cultured overnight. *Popu*CATH (25, 50, 100, or 200 µg/mL), PBS or LL-37 (25, 50, 100, or 200 µg/mL, positive control) was added and incubated at 37°C for 0.5 hr. The supernatant was collected for mast cell degranulation assay. (**E**) Complement activation assay. Mouse serum was treated with PBS, EGTA inhibitor (10 mM), zymosan (0.5 mg/mL), *Popu*CATH (25, 50, 100, and 200 µg/mL) at 37°C for 1 hr. C3a des-Arg was measured by ELISA. (**F, G**) In vivo acute toxicity assay. C57BL/6 mice (18–20 g, n = 6) were intraperitoneally injected with *Popu*CATH at dose of 10, 20 and 40 mg/kg. At 24 hr post injection, kidneys, livers, hearts and spleens were collected for H&E staining (**F**), scale bar: 100 µm. Serum was collected for the creatinine and alanine aminotransferase (ALT) measurement (**G**). *p < 0.05, **p < 0.01, ***p < 0.001, ns, not significant.

while human cathelicidin LL-37 (positive control) markedly induced mast cell degranulation as described previously (*Niyonsaba et al., 2001*).

The effect of *Popu*CATH on complement activation was determined by measurement of C3a after incubation of *Popu*CATH with mouse serum. *Popu*CATH didn't exhibit significant effect on complement activation at concentration up to 200 µg/mL, in contrast to the controls of ethylene glycol tetraacetic acid (EGTA) and zymosan (*Figure 3E*).

The in vivo acute toxicity of *Popu*CATH was assessed by histopathological study and blood routine examination at 24 hr after intraperitoneal injection of *Popu*CATH in C57BL/6 mice. Compared to PBS treatment, *Popu*CATH treatment did not cause pathological abnormality in kidney, liver, heart, and spleen at any dose tested up to 40 mg/kg (*Figure 3F*). The levels of alanine aminotransferase (ALT) and creatinine in the sera of mice between *Popu*CATH- and PBS-treated groups showed no significant

difference (*Figure 3G*), suggesting that high or low doses of *Popu*CATH does not affect the hepatic and renal function of mice.

The highest amount of administered substance that does not kill tested animals was recorded as the maximum tolerable dose (*Scott et al., 2007*). The maximum tolerable dose of *Popu*CATH was tested in C57BL/6 mice by intravenous and intraperitoneal delivery. The maximum tolerable dose of *Popu*CATH by intravenous delivery was between 75 and 100 mg/kg, and intraperitoneal delivery was between 125 and 150 mg/kg. Markedly, *Popu*CATH was not toxic at substantially higher concentrations via the intraperitoneal route, well above the doses (10, 20, and 40 mg/kg) used in the mouse models.

## *Popu*CATH provides prophylactic efficacy against bacterial infection in mice

We next examined whether *Popu*CATH protects against bacterial infection in mice as observed in tree frogs. Mice were intraperitoneally injected with *Popu*CATH at 8 or 4 hr (−8 or −4 hr) prior to, or 4 hr after ( + 4 hr) intraperitoneal inoculation of bacteria. Compared to PBS injection, *Popu*CATH injection at −8 or −4 hr significantly reduced the bacterial loads in the abdominal cavity of mice post Gram-negative bacteria (*E. coli*, *A. baumannii*) and Gram-positive bacteria (*S. aureus*, methicillin-resistant *S. aureus*, MRSA) inoculation (*Figure 4A*). Bacterial inoculation significantly elicited the production of pro-inflammatory cytokines (TNF-α, IL-1β, and IL-6) in mouse serum relative to control mice (sham), while *Popu*CATH injection at −8 or −4 hr significantly reduced the production of pro-inflammatory cytokines in mouse serum (*Figure 4B*). Consistent with these findings, *E. coli* or *S. aureus* inoculation markedly induced inflammatory damage in the lung, and *Popu*CATH injection at −4 or −8 hr obviously rescued this inflammatory damage induced by bacterial inoculation (*Figure 4C*), suggesting its prophylactic efficacy against bacterial infection, and *Popu*CATH (10 mg/kg) also showed prophylactic efficacy against bacterial infection via intravenous injection (*Figure 4—figure supplement 1*). However, at the dose of 10 mg/kg, *Popu*CATH treatment at +4 hr did not significantly reduce the bacterial loads in abdominal cavity, the production of pro-inflammatory cytokines in serum, and the inflammatory damage in lung as compared to control mice (sham, *Figure 4A–C*).

In order to further evaluate the prophylactic efficacy of *Popu*CATH against bacterial infection, mice were intraperitoneally injected with *Popu*CATH before a lethal dose of *E. coli* or MRSA inoculation, and the survival rates of mice were monitored for up to 7 days. Compared to PBS treatment, *Popu*CATH pretreatment markedly increased the survival rates of mice challenged by a lethal dose of *E. coli* or MRSA (*Figure 4D*). Besides, *Popu*CATH exhibited a better prophylactic efficacy than those of LL-37 (human cathelicidin) and IDR-1 (bovine cathelicidin derivative) against a lethal dose of bacterial infection (*Figure 4D*). We then evaluated the protective efficacy of *Popu*CATH in a CLP-induced sepsis model. We found that *Popu*CATH pretreatment markedly increased the survival rate of mice against CLP-induced sepsis (*Figure 4E*). These data suggested that *Popu*CATH (10 mg/kg) pretreatment effectively provided prophylactic efficacy against bacterial infection and prevented sepsis induced by a lethal dose of bacterial inoculation or CLP in mice.

## Intraperitoneal injection of *Popu*CATH induces leukocyte influx in both mice and tree frogs

Successful clearance of bacterial infection depended on an efficient phagocyte migration into the infectious sites (*Alves-Filho et al., 2010*; *Nathan, 2006*; *Scott et al., 2007*). We thereby investigated whether *Popu*CATH elicits phagocyte recruitment in mice. As shown in *Figure 5*, an intraperitoneal injection of *Popu*CATH (10 mg/kg) significantly induced the recruitment of leukocytes in the abdominal cavity (*Figure 5A&B*) and peripheral blood (*Figure 5C&D*) of mice, and *Popu*CATH was mainly chemotactic to myeloid cells with a negligible impact on lymphoid cells (*Figure 5*, *Figure 5—figure supplement 1*). In detail, the main myeloid cells recruited by *Popu*CATH were neutrophils, Ly6C^high macrophages and Ly6C^high monocytes in mouse abdominal cavity (*Figure 5A&B*), and were neutrophils, Ly6C^high monocytes and Ly6C^low monocytes in mouse peripheral blood (*Figure 5C&D*), indicating that intraperitoneal injection of *Popu*CATH significantly induced phagocyte influx in the abdominal cavity and peripheral blood of mice. Chemotactic kinetic assay indicated that the chemotactic effect induced by *Popu*CATH can last for 24 hr in abdominal cavity (*Figure 5—figure supplement 2*), and last for 48 hr in peripheral blood (*Figure 5—figure supplement 3*). Whereas an intraperitoneal injection

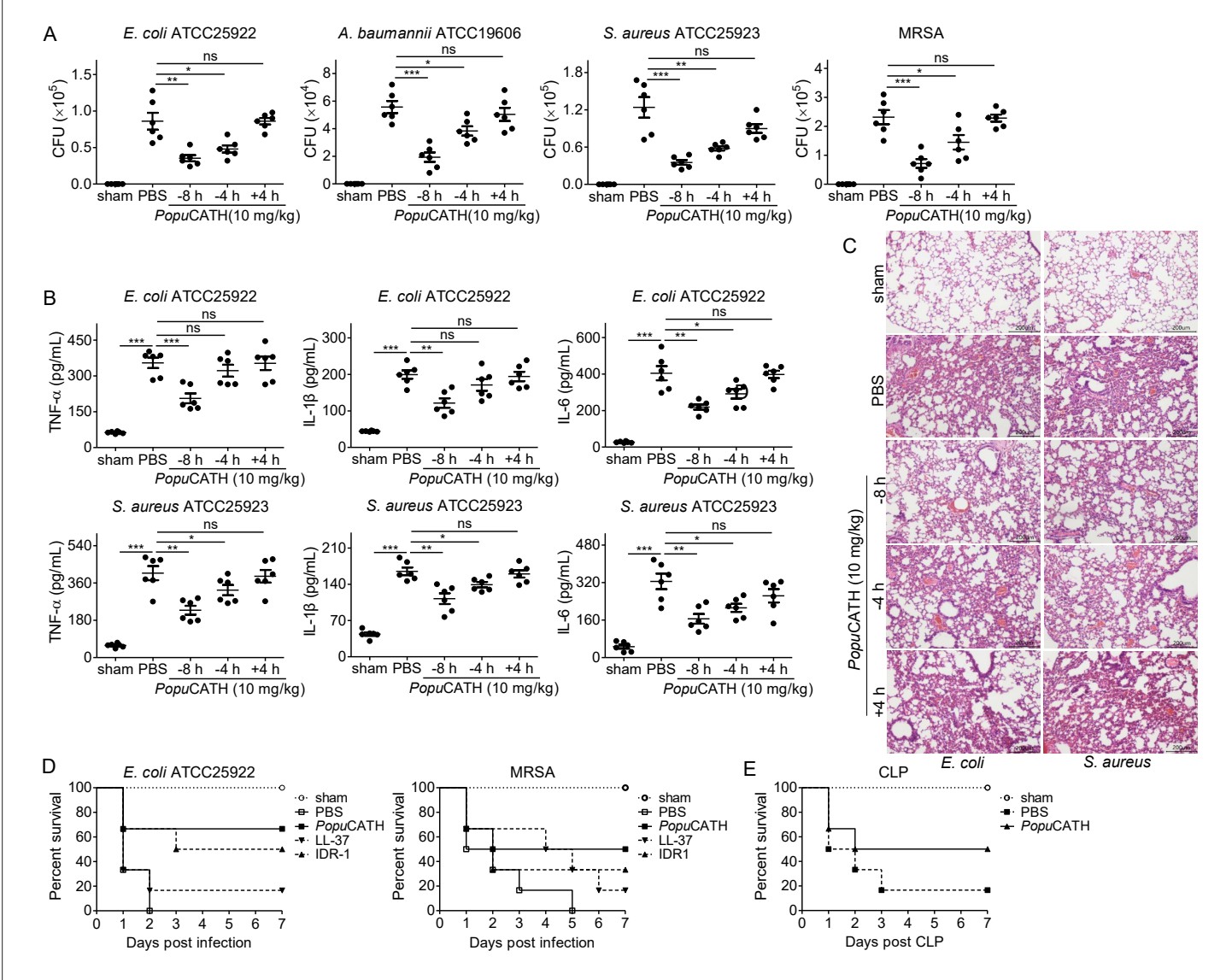

**Figure 4.** *Popu*CATH provides prophylactic efficacy against bacterial infection in mice. (**A–C**) *Popu*CATH (10 mg/kg) was intraperitoneally injected into C57BL/6 mice (18–20 g, n = 6) 8 or 4 hr prior to (–8 or –4 hr), or 4 hr after ( + 4 hr) *E. coli*, *A. baumannii*, *S. aureus* or methicillin-resistant *S. aureus* (MRSA) inoculation (2 × 10⁷ CFUs/mouse, intraperitoneal injection). At 18 hr post inoculation, peritoneal lavage was collected for bacterial load assay (**A**), serum was collected for cytokine assay (**B**), and lungs were taken for histopathological assay (**C**), scale bar: 200 μm. (**D**) C57BL/6 mice (18–20 g, n = 6) were intraperitoneally injected with *Popu*CATH (10 mg/kg), LL-37, or IDR-1 (control peptides) 4 hr prior to a lethal dose of *E. coli* (4 × 10⁷ CFUs/mouse) or MRSA (6 × 10⁸ CFUs/mouse, intraperitoneal injection) inoculation. The survival rates of mice were monitored for 7 days. (**E**) C57BL/6 mice (18–20 g, n = 6) were intraperitoneally injected with *Popu*CATH (10 mg/kg) at 8 and 4 hr (two times) prior to (–8 and –4 hr) CLP. At 0 hr, mice were anaesthesied with ketamine (100 mg/kg), and CLP was performed. The survival rates of mice were monitored for 7 days. *p < 0.05, **p < 0.01, ***p < 0.001, ns, not significant.

The online version of this article includes the following figure supplement(s) for figure 4:

**Figure supplement 1.** Prophylactic efficacy of *Popu*CATH against bacterial infection in mice by intravenous injection or intramuscular injection.

**Figure supplement 2.** Therapeutic efficacy of different doses of *Popu*CATH against bacterial infection in mice.

**Figure supplement 3.** Intraperitoneal injection of *Popu*CATH reduces the bacterial load in peripheral blood.

**Figure supplement 4.** Glycine, arginine, serine residues are key structural requirements for *Popu*CATH-mediated prophylactic efficacy against bacterial infection.

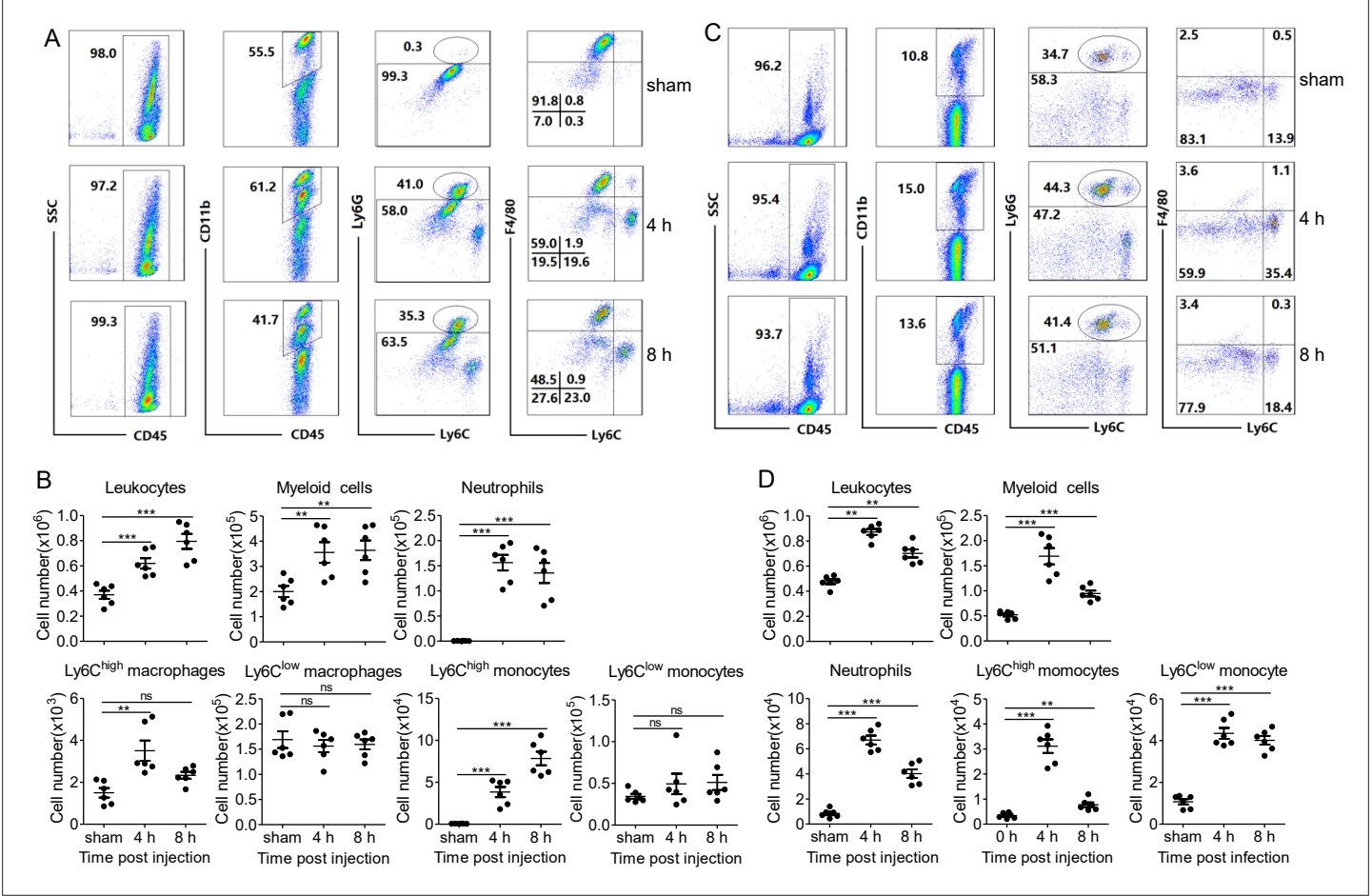

**Figure 5.** Intraperitoneal injection of *Popu*CATH induces leukocyte influx in mice. (**A, C**) Representative flow cytometry plots and proportions of major population of myeloid cells in peritoneal lavage (**A**) and peripheral blood (**C**) are shown. (**B, D**) Quantitative summary of leukocytes, myeloid cells, neutrophils, monocytes/macrophages in peritoneal lavage (**B**) and peripheral blood (**D**). C57BL/6 mice (18–20 g, n = 6) were intraperitoneally injected with *Popu*CATH (10 mg/kg) dissolved in 0.2 mL PBS. Sham mice received the same volumes of PBS. At 4 and 8 hr post injection, cells in peritoneal lavage and peripheral blood were collected for flow cytometry assay. **p < 0.01, ***p < 0.001, ns, not significant.

The online version of this article includes the following figure supplement(s) for figure 5:

**Figure supplement 1.** Intraperitoneal injection of *Popu*CATH does not induce T and B lymphocyte influx in mice.

**Figure supplement 2.** Phagocyte influx in abdominal cavity induced by *Popu*CATH can last for 24 hr.

**Figure supplement 3.** Phagocyte influx in peripheral blood induced by *Popu*CATH can last for 48 hr.

**Figure supplement 4.** Leukocyte influx induced by intraperitoneal injection of LPS in mice.

**Figure supplement 5.** Intraperitoneal injection of *Popu*CATH induces leukocyte influx in tree frogs.

of LPS (20 µg/mouse) elicited a different pattern of cellular influx in mice (*Figure 5—figure supplement 4*), suggesting that chemotaxis observed in mouse peritoneal cavity and peripheral blood were specifically due to *Popu*CATH rather than possible endotoxin contamination. In addition, we assayed if *Popu*CATH induce leukocyte influx in tree frogs. As shown in *Figure 5—figure supplement 5*, an intraperitoneal injection of *Popu*CATH obviously recruited leukocytes to the abdominal cavity of tree frogs, which is consistent with the data observed in mice.

## Neutrophils and monocytes/macrophages, but not T and B cells, are required for the protective efficacy of *Popu*CATH in mice

*Popu*CATH was chemotactic to neutrophils and monocytes/macrophages in both mouse abdominal cavity and peripheral blood. To examine whether the protective capacity of *Popu*CATH depends on these phagocytic cell types, we evaluated the prophylactic efficacy of *Popu*CATH in mice after

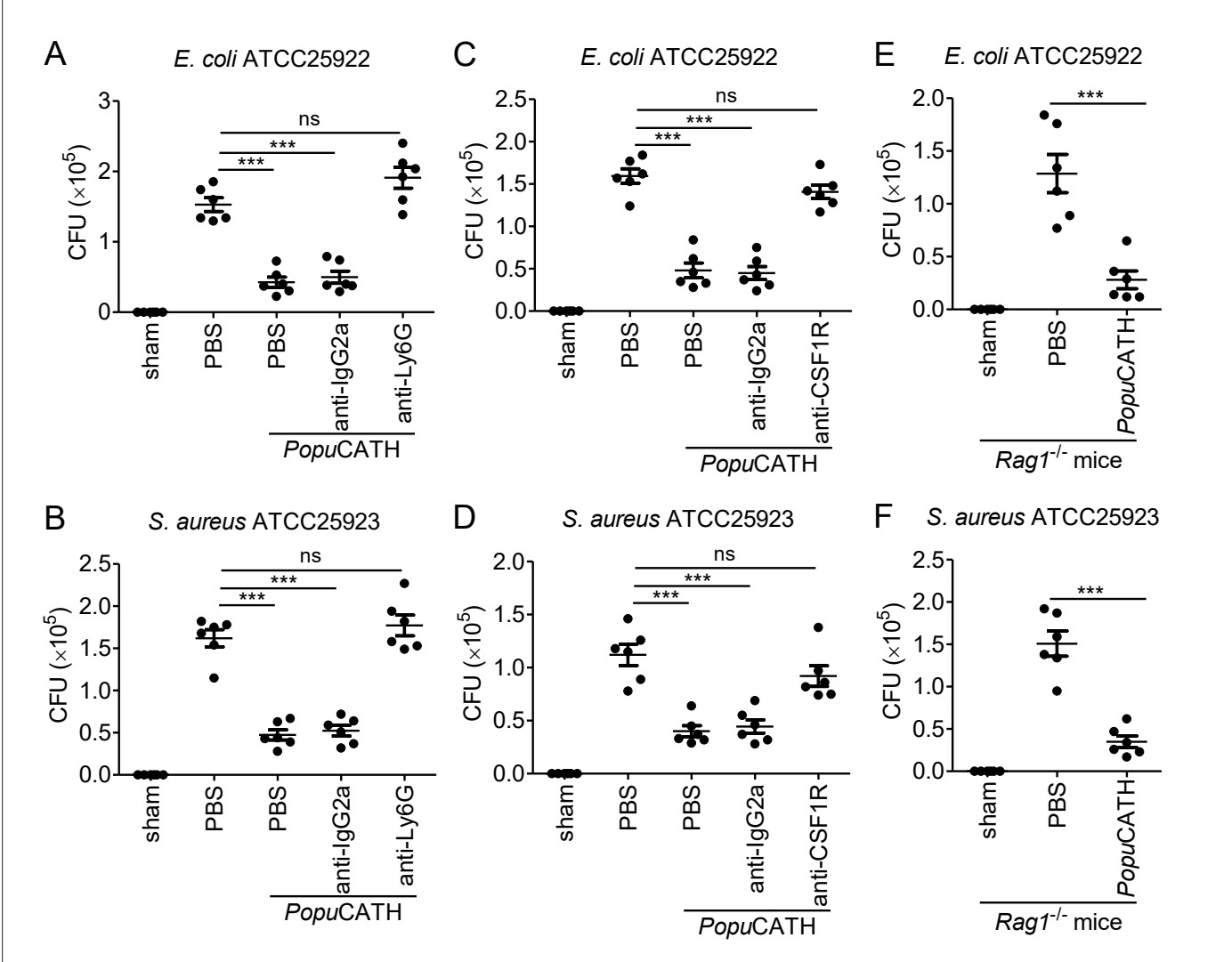

**Figure 6.** Neutrophils and monocytes/macrophages, but not T and B cells, are required for the prophylactic efficacy of *Popu*CATH in mice. (**A, B**) Protective efficacy of *Popu*CATH against *E. coli* (**A**) or *S. aureus* (**B**) in neutrophil depletion mice. Anti-Ly6G antibody or rat IgG2a isotype antibody were intraperitoneally injected into C57BL/6 mice (18–20 g, n = 6) at doses of 500 µg/mouse on day 0 and day 2, respectively. (**C, D**) Protective efficacy of *Popu*CATH against *E. coli* (**C**) or *S. aureus* (**D**) in monocyte/macrophage depletion mice. Anti-CSF1R antibody or rat IgG2a isotype antibody were intraperitoneally injected into C57BL/6 mice (18–20 g, n = 6) at doses of 1 mg/mouse on day 0 followed by 0.3 mg/mouse on day 1 and day 2, respectively. (**E, F**) Protective efficacy of *Popu*CATH against *E. coli* (**E**) or *S. aureus* (**F**) in *Rag1*[−/−] mice (18–20 g, n = 6). At 4 hr before *E. coli* or *S. aureus* (2 × 10⁷ CFUs/mouse) inoculation, *Popu*CATH (10 mg/kg) was intraperitoneally injected into neutrophil depletion mice (on day 3), monocyte/macrophage depletion mice (on day 3), and *Rag1*[−/−] mice. At 18 hr post bacterial inoculation, peritoneal lavage was collected for the bacterial load assay. ***p < 0.001, ns, not significant.

The online version of this article includes the following figure supplement(s) for figure 6:

**Figure supplement 1.** Scavenging efficiency of neutrophils and monocytes/macrophages by anti-Ly6G and anti-CSF1R antibody.

the neutrophils or monocytes/macrophages were depleted by anti-Ly6G or anti-CSF1R antibody (***Figure 6—figure supplement 1***). As shown in ***Figure 6***, *Popu*CATH failed to provide prophylactic efficacy against *E. coli* (***Figure 6A***) and *S. aureus* (***Figure 6B***) infection in neutrophil-depleted mice, and *Popu*CATH was not efficacious against *E. coli* (***Figure 6C***) and *S. aureus* (***Figure 6D***) infection in monocyte/macrophage-depleted mice. As mentioned above, *Popu*CATH was primarily chemotactic to myeloid cells with a negligible impact on lymphoid cells. To confirm this finding, we next tested its prophylactic efficacy in *Rag1*[−/−] mice, which are T and B lymphocyte-deficient mice. In contrast to neutrophil and monocyte/macrophage depletion, *Popu*CATH still provided prophylactic efficacy

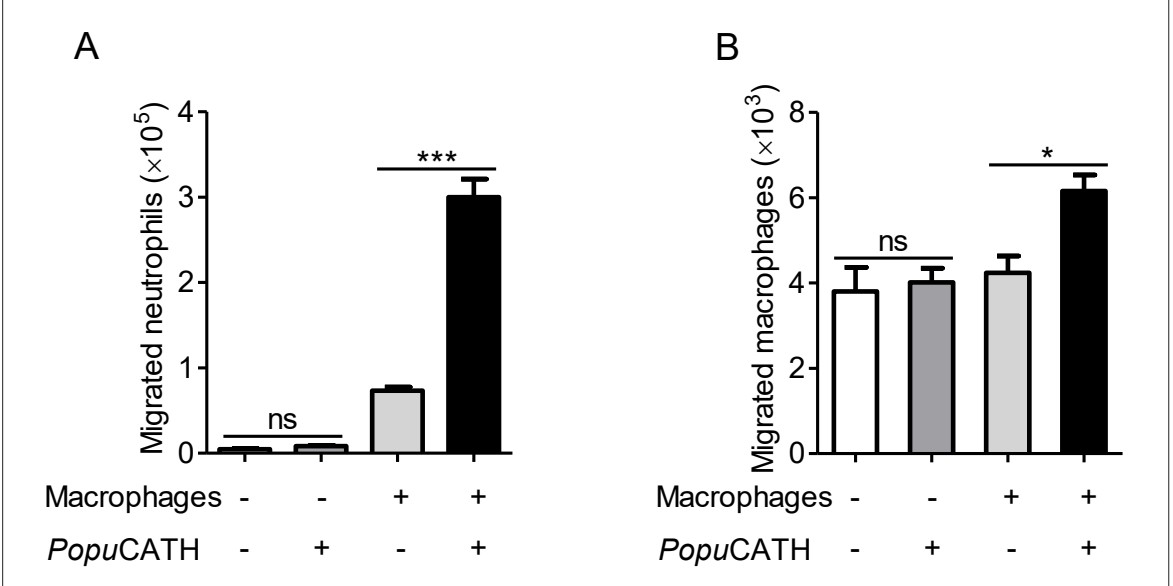

**Figure 7.** *Popu*CATH-induced phagocyte migration relies on its effect on macrophages. For the direct chemotactic effect of *Popu*CATH to neutrophils or macrophages, 100 µL of neutrophil suspension (**A**) or macrophage suspension (**B**) (5 × 10⁶ cells/mL) was added to the upper chamber, and 500 µL of *Popu*CATH (10 µM, dissolved in medium) or medium was added to the lower chamber. After neutrophils and macrophages were migrated at 37 ℃ for 8 hr, the increased cells in the lower chamber were collected and counted using a hemocytometer. For the co-cultured system, 500 µL of macrophage suspension (5 × 10⁶ cells/mL) was seeded in the lower chamber. After macrophages were adherent to the lower chamber, 100 µL of neutrophil suspension (**A**) or macrophage suspension (**B**) (5 × 10⁶ cells/mL) was added to the upper chamber. Then, the medium in the lower chamber was replaced with 500 µL of *Popu*CATH (10 µM, dissolved in medium) or fresh medium. Neutrophils and macrophages were migrated at 37 ℃ for 8 hr. The reduced cells in the upper chamber were counted using a hemocytometer. *p< 0.05, ***p < 0.001, ns, not significant.

against *E. coli* (*Figure 6E*) and *S. aureus* (*Figure 6F*) infection in *Rag1⁻/⁻* mice. These data suggested that myeloid cells, but not lymphoid cells, are required for *Popu*CATH-mediated protection against bacterial infection in mice.

## *Popu*CATH-induced phagocyte migration relies on its effect on macrophages

Given the increase in neutrophils and monocytes/macrophages in the abdominal cavity and peripheral blood, we were interested to investigate if *Popu*CATH acts as a chemoattractant for neutrophils and macrophages. As shown in *Figure 7*, *Popu*CATH (10 µM) did not directly induce neutrophil migration (*Figure 7A*) and macrophage migration (*Figure 7B*), suggesting that *Popu*CATH cannot act as a chemo-attractant for neutrophils and macrophages. Macrophages have been shown to produce chemokines/cytokines that recruit other cells, and macrophages are the major immune cells in mouse abdominal cavity (*Scott et al., 2007*; *Yang et al., 2021*). We next investigated whether *Popu*CATH induce phago-cyte migration in the presence of macrophages. As shown in *Figure 7*, *Popu*CATH (10 µM) markedly induced neutrophil migration (*Figure 7A*) and macrophage migration (*Figure 7B*) in the presence of peritoneal macrophages. The addition of *Popu*CATH (10 µM) in the lower chamber elicited about 2.3 × 10⁵ neutrophil migration (*Figure 7A*) and 2.0 × 10³ macrophage migration (*Figure 7B*) when perito-neal macrophages were cultured in the lower chamber, implying that *Popu*CATH-induced phagocyte migration might rely on *Popu*CATH-triggered immune response in macrophages.

## *Popu*CATH selectively induced the production of chemokines/cytokines in macrophages and mice

To confirm whether *Popu*CATH-induced phagocyte migration relies on *Popu*CATH-triggered immune response in macrophages, we stimulated mouse peritoneal macrophages with a single dose of *Popu*CATH (10 µM) for 4 hr and analysed the mRNA levels of chemokines/cytokines. As shown in *Figure 8A*, the mRNA levels of *Cxcl1, Cxcl2, Cxcl3, Il1b,* and *Il6* were significantly increased by 60.9-, 74.2-, 17.0-, 15.5-, and 18.6-fold in peritoneal macrophages post *Popu*CATH treatment relative to PBS

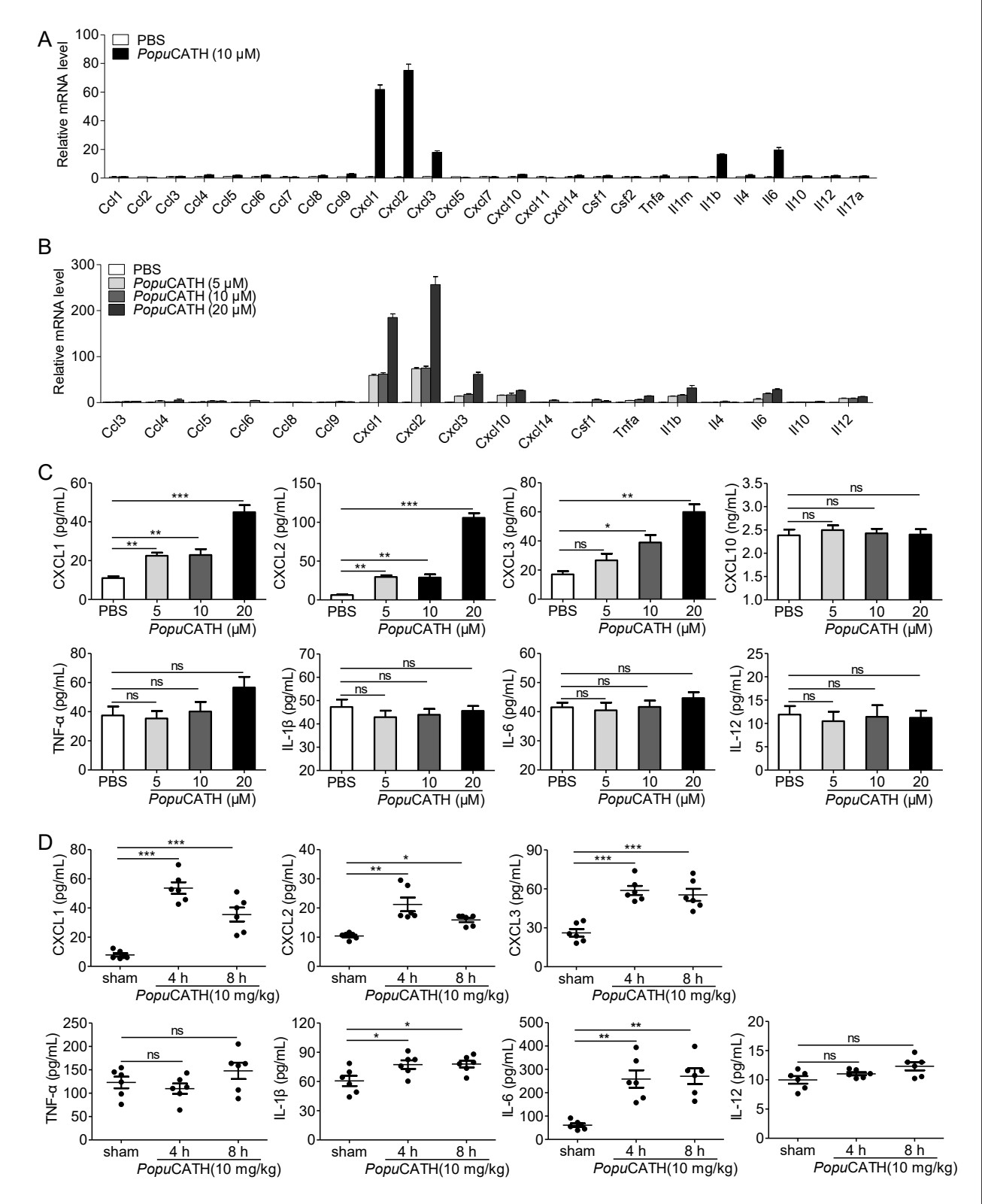

**Figure 8.** *Popu*CATH selectively induced the production of chemokines in macrophages and mice. (**A**) The mRNA levels of chemokines/cytokines in macrophages induced by *Popu*CATH (10 μM). (**B**) Verification of the upregulated chemokines/cytokines observed in panel A by qPCR. (**C**) The protein levels of chemokine/cytokine production in macrophages induced by *Popu*CATH (10 μM). Macrophages (5 × 10⁵ cells/well, in 2% FBS DMEM) were seeded in 24-well plates, and a single dose of *Popu*CATH (10 μM, dissolved in PBS) (**A**) or different doses of *Popu*CATH (5, 10, and 20 μM, dissolved

*Figure 8 continued on next page*

*Figure 8 continued*

in PBS) (**B**), or PBS was added. After incubation at 37°C for 4 hr, cells were collected, and the mRNA levels of chemokines/cytokines were detected by qPCR analysis, the protein levels of chemokines/cytokines were quantified by ELISA (**C**). (**D**) The protein levels of chemokine/cytokines in mice induced by *Popu*CATH (10 mg/kg). C57BL/6 mice (18–20 g, n = 6) were intraperitoneally injected with *Popu*CATH (10 mg/kg) dissolved in 0.2 mL PBS. Sham mice received the same volumes of PBS. At 4 and 8 hr post injection, peritoneal lavage was collected for quantification of the protein levels of chemokines/cytokines by ELISA. *p < 0.05, **p < 0.01, ***p < 0.001, ns, not significant.

The online version of this article includes the following figure supplement(s) for figure 8:

**Figure supplement 1.** Flow cytometry analysis of mouse peritoneal macrophages.

**Figure supplement 2.** *Popu*CATH does not promote macrophage phagocytosis.

**Figure supplement 3.** The effect of *Popu*CATH on LTA- or LPS-induced inflammation in macrophages.

treatment (p < 0.05). In contrast, the mRNA levels of the other chemokines and cytokines, including *Ccl4*, *Ccl5*, *Ccl6*, *Ccl8*, *Ccl9*, *Ccl10*, *Ccl14*, *Csf1*, *Il4*, *Il12*, and *Tnfa* were slightly upregulated in peritoneal macrophages post *Popu*CATH treatment relative to PBS treatment, ranging from 1.5- to 3-fold (p > 0.05). We next stimulated mouse peritoneal macrophages with different dose of *Popu*CATH (5, 10, and 20 μM) to verify the results observed in *Figure 8A*. As shown in *Figure 8B*, mRNA levels of chemokines (*Cxcl1*, *Cxcl2*, and *Cxcl3*) and cytokines (*Il1b* and *Il6*) were significantly upregulated in a dose-dependent manner (p < 0.05). The others didn't generate a dose-dependent effect. To confirm the results observed by mRNA quantification, we detected the protein levels of the upregulated chemokines/cytokines by ELISA. *Popu*CATH significantly induced the protein production of CXCL1, CXCL2, and CXCL3 in a dose-dependent manner, whereas *Popu*CATH did not significantly induce the protein production of CXCL1, TNF-α, IL-1β, and IL-6 although their mRNA levels were upregulated (*Figure 8C*). In vivo assay showed that an intraperitoneal injection of *Popu*CATH (10 mg/kg) significantly induced the production of the chemokines (CXCL1, CXCL2, and CXCL3) as well as the pro-inflammatory cytokines (IL-1β and IL-6) in mouse abdominal cavity (*Figure 8D*). The results indicated that *Popu*CATH directly acted on macrophages and selectively induced the production of chemoattractant which are critical for the recruitment of phagocytes.

### *Popu*CATH-induced chemokine production in macrophages were partially dependent on p38/ERK MAPKs and NF-κB signaling pathways

To investigate the signaling pathways by which chemokines were induced by *Popu*CATH in macrophages, mouse peritoneal macrophages were pretreated with various inhibitors, including p38, ERK1/2, JNK1/2, PI3K and NF-κB, and responses induced by *Popu*CATH were analysed. As shown in *Figure 9A*, chemokines (CXCL1, CXCL2, and CXCL3) induced by *Popu*CATH were markedly attenuated after p38/ERK MAPKs, or NF-κB blockade, whereas inhibitors of JNK MAPK and PI3K pathway had no significant effect on *Popu*CATH-induced chemokine production in macrophages. Consistent with these results, *Popu*CATH (10 μM) significantly activated p38/ERK MAPKs and NF-κB p65 (*Figure 9B&C*). But inhibition of p38/ERK MAPKs or NF-κB signaling pathways did not completely blocked *Popu*CATH-mediated chemokine production in macrophages, we cannot not exclude other possible signaling pathways were involved. The data suggested that *Popu*CATH-mediated chemokine production in macrophages partially depended on p38/ERK MAPKs and NF-κB signaling pathways.

### *Popu*CATH promoted neutrophil phagocytosis through eliciting neutrophil extracellular traps

It is noted that *Popu*CATH primarily drove neutrophil influx in both peritoneal cavity and peripheral blood, and peaked at 4 hr post intraperitoneal injection of *Popu*CATH with an increment of approximately $1.54 \times 10^5$ neutrophils in mouse abdominal cavity (*Figure 5A&B*) and $5.64 \times 10^6$ neutrophils in mouse peripheral blood (*Figure 5C&D*) relative to control mice (sham), indicating that neutrophils exhibited a rapid response to *Popu*CATH. We herein tried to understand whether *Popu*CATH directly act on neutrophils to promote bacterial clearance. As illustrated in *Figure 10A*, *Popu*CATH significantly promoted phagocytic uptake of bacterial particles by mouse neutrophils. To investigate the mechanism by which *Popu*CATH promoted the phagocytic activity of neutrophils, the capacity of *Popu*CATH to induce neutrophil extracellular traps (NETs) were detected as indicated in *Figure 10B*. Single treatment of *Popu*CATH or PMA (positive control) markedly induced the formation of NETs as

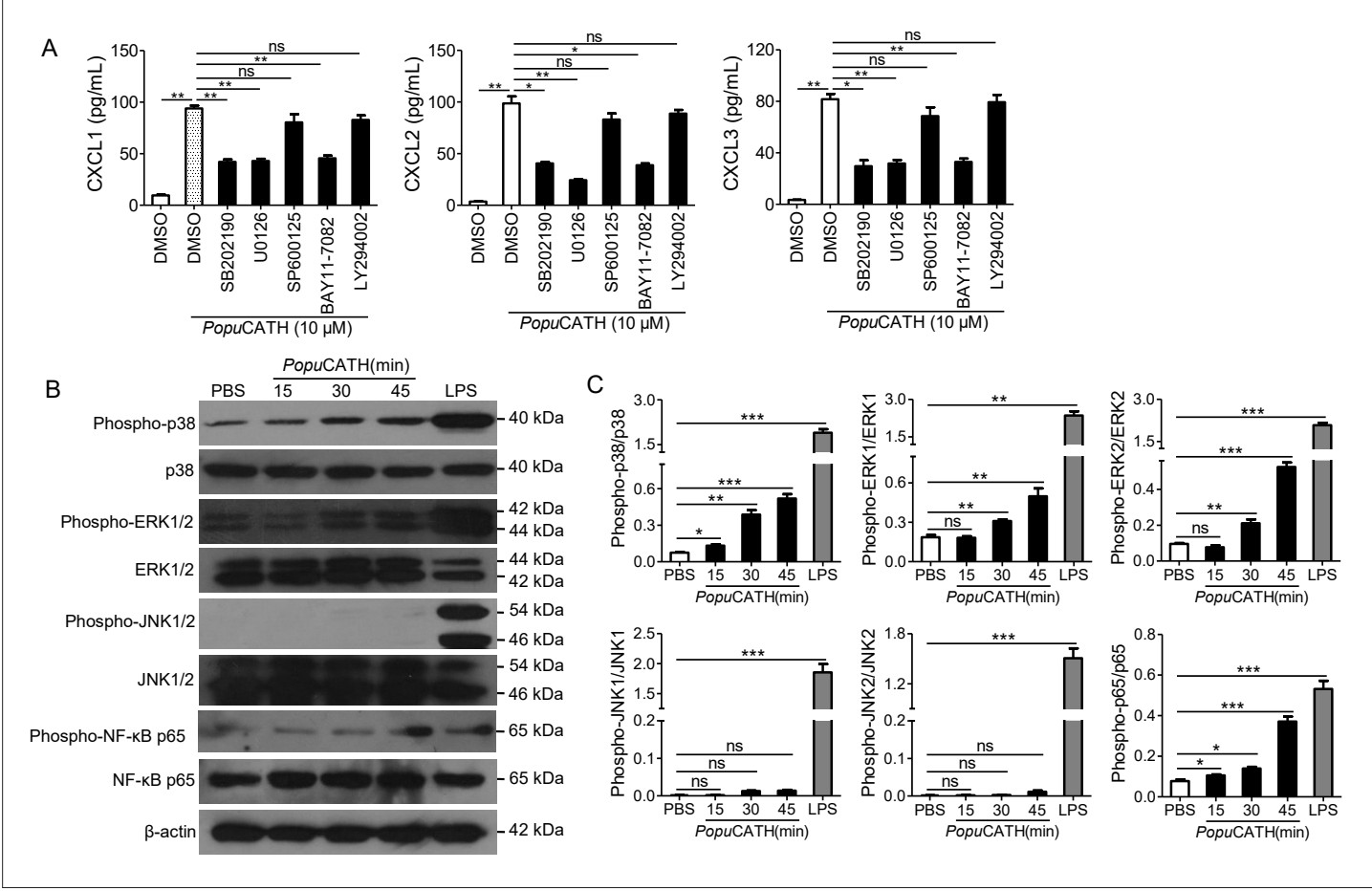

**Figure 9.** *Popu*CATH-induced chemokine production in macrophages were partially dependent on p38/ERK MAPKs and NF-$\kappa$B signaling pathways. (**A**) Effects of MAPK, PI3K, and NF-$\kappa$B inhibitors on *Popu*CATH-induced chemokine production in macrophages. Macrophages ($5 \times 10^5$ cells/well, in 2% FBS DMEM) were seeded in 24-well plates. The adherent macrophages were pre-incubated with p38 inhibitor (SB202190, 10 μM), ERK inhibitor (U0126, 10 μM), JNK inhibitor (SP600125, 10 μM), NF-$\kappa$B inhibitor (BAY11-7082, 2 μM), or PI3K inhibitor (LY294002, 10 μM) for 1 hr, respectively. Then, cells were stimulated with *Popu*CATH (10 μM) for 4 hr. The protein levels of chemokines were quantified by ELISA. (**B**) Western blot analysis of the effects of *Popu*CATH on MAPKs and NF-$\kappa$B. Macrophages ($2 \times 10^6$ cells/well, in 2% FBS DMEM) were seeded in 6-well plates. *Popu*CATH (10 μM) was added and incubated at 37°C for 15, 30, and 45 min, respectively. LPS (100 ng/mL, positive control) or PBS (solvent of peptide) was added and incubated for 30 min. The cells were collected for western blot analysis. (**C**) Ratio analysis. The ratios of phosphorylated-p38, JNK, ERK, and NF-$\kappa$B p65 to total p38, JNK, ERK, and NF-$\kappa$B p65 were assayed by image J, respectively. *p < 0.05, **p < 0.01, ***p < 0.001, ns, not significant.

The online version of this article includes the following source data for figure 9:

**Source data 1.** The original images of the unedited blots and images with the uncropped blots with the relevant bands clearly labelled.

compared to PBS treatment, and co-treatment of *Popu*CATH and PMA increased the formation of NETs relative to single *Popu*CATH or PMA treatment, indicating that *Popu*CATH-enhanced neutrophil phagocytosis is attributed to *Popu*CATH-induced NETs formation.

## Discussion

We herein identified a novel amphibian cathelicidin designated as *Popu*CATH. In vitro antimicrobial assay, *Popu*CATH was devoid of any antimicrobial activity, including no significant effects on bacterial growth, metabolic activity and surface morphology, which is different from the bactericidal amphibian cathelicidins in previous studies. Intriguingly, although *Popu*CATH lacks direct antibacterial activities in vitro, it effectively provides prophylactic efficacy against bacterial infection in vivo with a broad spectrum, including Gran-negative bacteria, Gram-positive bacteria, and even clinically isolated methicillin-resistant *S. aureus*. Intraperitoneal injection of *Popu*CATH before bacterial

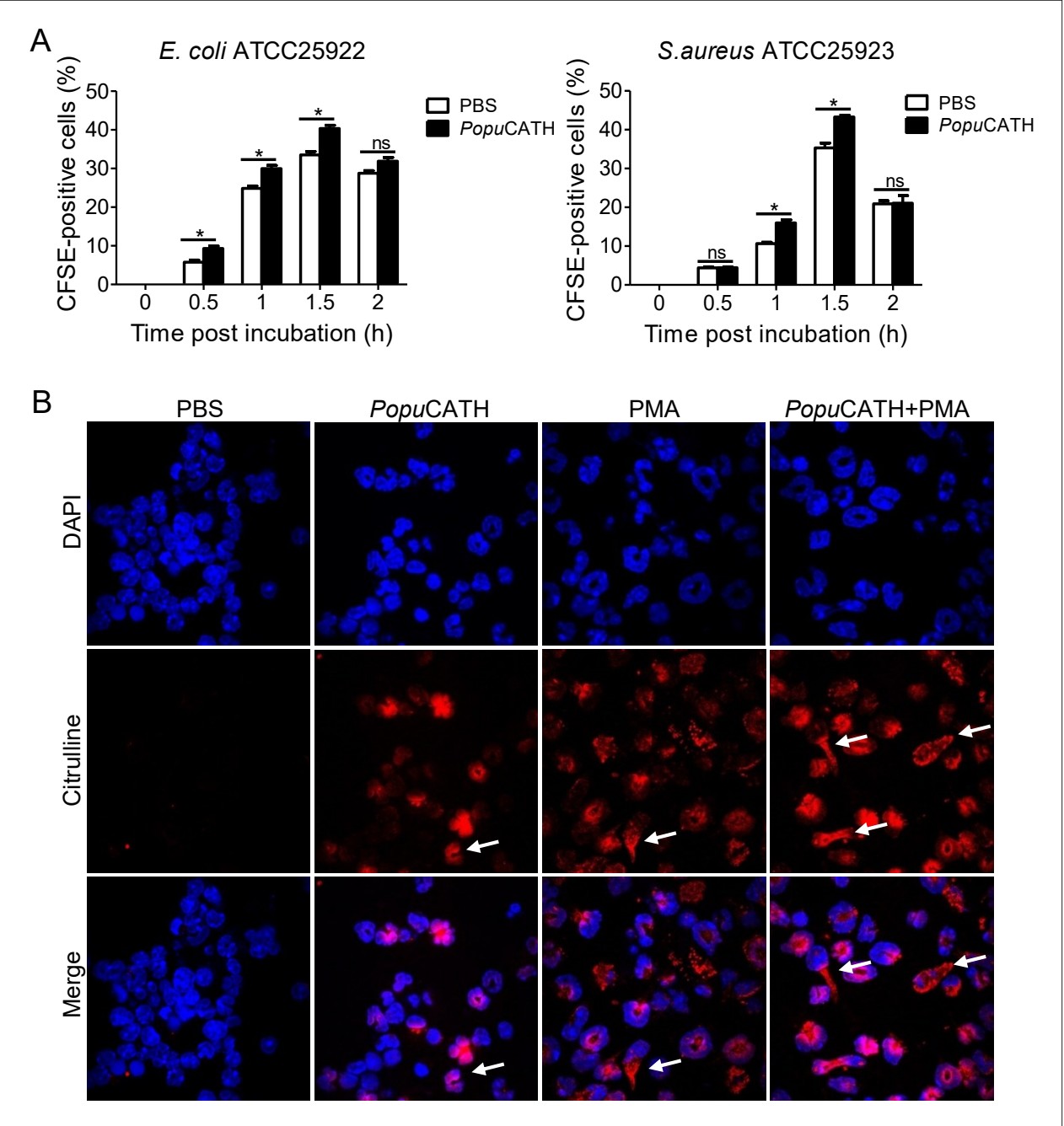

**Figure 10.** *Popu*CATH promoted neutrophil phagocytosis through enhancing neutrophil extracellular traps formation. (**A**) Enhancement of neutrophil phagocytosis of *S. aureus* and *E. coli* by *Popu*CATH. Neutrophils were pre-incubated with *Popu*CATH (10 μM) or PBS (solvent of peptide) for 1 hr, and CFSE-labelled bacterial particles were added and incubated for indicated time points. The CFSE fluorescence were analysed by flow cytometry as a measure of the phagocytic uptake of the bacterial particles. (**B**) Enhancement of neutrophil extracellular formation. Neutrophils were incubated with PBS, *Popu*CATH (10 μM), PMA (100 nM) or *Popu*CATH (10 μM)+ PMA (100 nM) at 37℃ for 4 hr, respectively. Nuclei and NETs were stained with DAPI (blue) or anti-Cit-H3 (red), respectively. NETs were observed using a confocal microscope ( × 60). *p < 0.05, ns, not significant.

The online version of this article includes the following figure supplement(s) for figure 10:

**Figure supplement 1.** Flow cytometry analysis of mouse bone-marrow-derived neutrophils.

**Figure supplement 2.** *Popu*CATH does not increase ROS production in neutrophils.

inoculation significantly attenuated the bacterial load in tree frogs and mice, reduced inflammatory responses induced by bacterial inoculation in mice, and increased the survival rates of septic mice induced by lethal dose of bacterial inoculation and CLP. Except for the intraperitoneal injection, intravenous injection of *Popu*CATH also effectively provides prophylactic efficacy against bacterial infection (*Figure 4—figure supplement 1*). The results indicates that *Popu*CATH can provide preventive capacity via both intraperitoneal and intravenous injection routes. While intramuscular injection had no significant preventive effects against bacterial infection (*Figure 4—figure supplement 1*). It is more likely that muscle is not rich in neutrophils, monocytes/macrophages, which are the key effector cells of *Popu*CATH. At the dose of 10 mg/kg, *Popu*CATH did not exhibit therapeutic efficacy against bacterial infection. In order to evaluate whether it has therapeutic efficacy at high doses, we increased the dose of *Popu*CATH. At doses of 20 and 40 mg/kg, *Popu*CATH significantly reduced the bacterial load when it was given at 4 hr after *E. coli* inoculation (*Figure 4—figure supplement 2*). It is possibly that bacteria have colonised in mice at 4 hr after bacterial inoculation, which need a higher dose of *Popu*CATH to drive more phagocytes for bacterial clearance. To the best of our knowledge, this is the first report of a non-bactericidal cathelicidin that can protect bacterial infection in vivo.

The extent of microbial infection-mediated host damage largely depends on host's immune status. If the host can effectively initiate an immune defense, the invading microbes will be cleared, and host damage induced by microbial infection will be prevented or controlled. On the contrary, if the host cannot effectively initiate an immune defense, host will lose a balanced protection and microbial infection-mediated host damage will follow (*Silva, 2010*). Neutrophils and macrophages are two professional phagocytic cell types, which comprise a myeloid phagocyte system of host. Neutrophils and macrophages usually work together in innate immunity as complementary partners of the myeloid phagocytic system. The local and global distribution patterns of neutrophils and macrophages are key immune parameter of host, which play critical roles in initiating effective immune defense against invading microbes. Our findings revealed that intraperitoneal injection of *Popu*CATH effectively elicited neutrophil and monocyte/macrophage influx in mice, and depletion of neutrophils or monocytes/macrophages blocked *Popu*CATH-mediated protection, indicating that *Popu*CATH-mediated protection depends on *Popu*CATH-induced neutrophil and monocyte/macrophage influx. Previous investigations have demonstrated that successful clearance of invading microbes largely depends on efficient migration of these cell types into the infectious sites (*Alves-Filho et al., 2010*; *Li et al., 2013*; *Nathan, 2006*; *Scott et al., 2007*). These suggest that intraperitoneal injection of *Popu*CATH enhanced the myeloid phagocytic system of mice, thus providing prophylactic efficacy against bacterial infection.

Macrophages have been shown to phagocytose and directly kill bacteria (*Nijnik et al., 2010*; *Scott et al., 2007*). In our study, we found that *Popu*CATH did not promote in vitro phagocytosis of fluorescently labelled bacterial particles by mouse peritoneal macrophages (*Figure 8—figure supplement 2*), suggesting no direct stimulation of the phagocytic activity of macrophages by *Popu*CATH. But *Popu*CATH significantly exhibited immunomodulatory effects on macrophages to induce phagocyte influx. In addition, neutrophils are principal phagocytes in the innate defense system and kill pathogens through mechanisms like oxidative killing activity and release of neutrophil extracellular traps (*Neumann et al., 2014*; *Niyonsaba et al., 2013*; *Rowe-Magnus et al., 2019*), and an influx of neutrophils to the site of infection is pivotal for the clearance of infectious bacteria (*Alves-Filho et al., 2010*). In our study, *Popu*CATH was merely demonstrated to promote neutrophil phagocytosis through inducing NET formation, but not significantly elicited oxidative killing activity of neutrophils (*Figure 10—figure supplement 2*). These results indicated that macrophages and neutrophils responded to *Popu*CATH in their own manner.

The expression profiles of chemokines/cytokines in vitro and in vivo are somewhat different, but we observed that the profiles of the major chemokines/cytokines induced by PopuCATH, such as CXCL1, CXCL2, and CXCL3, are similar. We presumed that PopuCATH-induced chemokines/cytokines in vivo are consumed timely. In addition, macrophage is the unique effector cell type of PopuCATH in vitro. While there are many other cell types in vivo, such as monocytes and neutrophils, and we cannot exclude these cells are responsive to *Popu*CATH and subsequently produce chemokines/cytokines. These may explain the subtle differences of *Popu*CATH-mediated chemokine/cytokine production in macrophages and mice. In mouse model, the production of chemokines/cytokines in mouse abdominal cavity peaked at 4 hr post injection of *Popu*CATH. The dynamic of CXCL1, CXCL2, and CXCL3

production is consistent with the dynamic of neutrophil, monocyte/macrophage recruitment in the mouse abdominal cavity and peripheral blood. Although the pretreatment with *Popu*CATH significantly induced the production of chemokines (CXCL1, CXCL2, and CXCL3) as well as pro-inflammatory cytokines (IL-1β and IL-6), *Popu*CATH ultimately attenuated the inflammatory response by decrease of TNF-α, IL-1β, and IL-6 levels post Gram-negative and Gram-positive bacterial infection. Cathelicidins are able to block Toll-like receptor (TLR)-mediated inflammatory responses, including those mediated by TLR2 and TLR4 (*Coorens et al., 2017*; *Mookherjee et al., 2006*; *Wei et al., 2013*). In this study, *Popu*CATH did not affect LTA- and LPS-stimulated inflammatory responses in mouse peritoneal macrophages (*Figure 8—figure supplement 3*), suggesting that the anti-inflammatory effects of *Popu*CATH are independent of TLRs, and the attenuation of the inflammation may be secondary to the decrease of bacterial growth.

Some of the properties of *Popu*CATH are reminiscent of the activities of other cathelicidins like LL-37 (*Chen et al., 2000*), CRAMP (*Kurosaka et al., 2005*), and OH-CATH30 (*Li et al., 2013*), which selectively modulated innate immune responses and have been proposed to mediate protection in animal models. Compared to theses bactericidal cathelicidins with immunomodulatory properties, (i) the usage of *Popu*CATH is unlikely to induce drug-resistance because the peptide is unable to directly elicit stress on microbes. (ii) *Popu*CATH showed low side effects unlike LL-37 (*Bąbolewska and Brzezińska-Błaszczyk, 2015*). (iii) The expression profile of chemokines/cytokines in response to *Popu*CATH were largely different from those of other cathelicidins. (iv) Key effector cells for *Popu*CATH were also largely different from those of LL-37, CRAMP and OH-CATH30. For example, human cathelicidin peptide LL-37 has been shown to directly recruit neutrophils, monocytes, mast cells, and T lymphocytes (*Chen et al., 2000*; *Niyonsaba et al., 2002*). While *Popu*CATH did not directly recruited leukocytes, it just recruited neutrophils and monocytes/macrophages via inducing chemokine/cytokine production in macrophages. In addition, *Popu*CATH just elicited neutrophil and monocyte/macrophage recruitment, but not T and B lymphocytes. Intraperitoneal injection of *Popu*CATH significantly drove phagocyte influx in both abdominal cavity and peripheral blood, demonstrating that it effectively regulated both local and global innate immune response. As a result, *Popu*CATH pretreatment effectively reduced the bacterial load in both abdominal cavity and peripheral blood (*Figure 4—figure supplement 3*). (v) Intriguingly, *Popu*CATH is a glycine-rich cathelicidin containing 21 glycine residues. The amino acid component is different from LL-37, CRAMP, and OH-CATH30, which are not special residue-rich cathelicidins. The substitution of glycine residues of *Popu*CATH with alanine residues significantly resulted in a reduced efficacy against bacterial infection (*Figure 4—figure supplement 4*). In addition, *Popu*CATH contains 10 arginine residues and seven serine residues (*Supplementary file 1*). The substitution of arginine residues or serine residues with alanine residues also significantly led to a decreased efficacy against bacterial infection (*Figure 4—figure supplement 4*). These data demonstrated that these enriched amino acid residues, including 21 glycine residues, 10 arginine residues, and 7 serine residues, are key structural requirements for *Popu*CATH-mediated protective efficacy against bacterial infection, and *Popu*CATH-mediated protection were specifically due to its unique structure. The first frog-derived cathelicidin is also rich in glycine residues. But it has different amino acid sequence with *Popu*CATH and exhibits direct antibacterial activity unlike *Popu*CATH (*Hao et al., 2012*). Cathelicidin antimicrobial peptides display a high structural diversity, and the diverse structures are responsible for their diverse functions. Accordingly, it is not difficult to understand that these two frog cathelicidins have different functions against bacteria.

Recently, many progresses have been achieved in the development of anti-resistance therapy for combatting multidrug resistant bacterial infection. Pre-clinical and clinical data pointed out host-directed therapeutic approaches to enhance 'pauci-inflammatory' microbial killing in myeloid phagocytes merited particular attention (*Watson et al., 2020*). Host-based therapeutic strategies can maximise microbial clearance and minimise host's harmful consequences induced by inflammatory response, which has great promise. *Popu*CATH did not show any direct effects on bacteria, but effectively prevented bacterial infection through eliciting phagocyte influx and slightly promoting neutrophil phagocytosis. *Popu*CATH-mediated protection against bacterial infection can be considered as a classic host-based therapeutic strategy, and the non-bactericidal nature of *Popu*CATH may reduce the selective pressures that drive bacterial resistance. In an era of emerging and re-emerging infectious diseases, discovery and development of naturally occurring non-bactericidal antimicrobial peptides like *Popu*CATH may facilitate us to prevent and overcome multidrug-resistant bacterial infection.

In summary, a glycine-rich amphibian cathelicidin, *Popu*CATH, was identified from tree frog. *Popu*CATH didn't show any direct effects on bacteria but provided protection against bacterial infection in vivo. PopuCATH acted as an immune defense regulator against bacterial infection by selective modulation of innate immune response. Our findings provide new insights into the development of non-bactericidal cathelicidins to prevent bacterial infection.

# Materials and methods

**Key resources table**

| Reagent type (species) or resource | Designation | Source or reference | Identifiers | Additional information |
|---|---|---|---|---|
| Cell line (*Homo sapiens*) | THP-1 | National Collection of Authenticated Cell Cultures (https://www.cellbank.org.cn/) | CSTR:19375.09.3101HUMSCSP567 | |
| Cell line (*Rattus norvegicus*) | RBL-2H3 | National Collection of Authenticated Cell Cultures (https://www.cellbank.org.cn/) | CSTR:19375.09.3101RATTCR7 | |
| Cell line (*Mus musculus*) | Macrophage | Peritoneal macrophages from C57BL/6 mice | | A primary cell line identified by flow cytometry |
| Cell line (*Mus musculus*) | Neutrophil | Bone marrow-derived neutrophils from C57BL/6 mice | | A primary cell line identified by flow cytometry |
| Commercial assay or kit | SMART cDNA Library Construction Kit | Clontech | Cat#: 634,901 | |
| Commercial assay or kit | Cell Counting Kit-8 | Dojindo | Cat#: CK04-500T | |
| Commercial assay or kit | Mouse C3a ELISA Kit | Wuhan Fine Biotech Co., Ltd | Cat#: EM0882 | |
| Commercial assay or kit | Alanine aminotransferase Assay Kit | Nanjing Jiancheng Bioengineering Institute | Cat#: C009-2-1 | |
| Commercial assay or kit | Creatinine Assay Kit | Nanjing Jiancheng Bioengineering Institute | Cat#: C011-2-1 | |
| Commercial assay or kit | Wright-Giemsa stain solution | Solarbio Life Sciences | Cat#: G1020 | |
| Commercial assay or kit | Trizol reagent | Life Technologies | Cat#: 15596018 | |
| Commercial assay or kit | PrimeScript RT reagent kit | Takara | Cat#: RR037A | |
| Commercial assay or kit | Mouse TNF-α ELISA Kit | eBioscience | Cat#: 88-7324-88, RRID:AB_2575080 | |
| Commercial assay or kit | Mouse IL-1β ELISA Kit | eBioscience | Cat#: 88-7013-88, RRID:AB_2574946 | |
| Commercial assay or kit | Mouse IL-6 ELISA Kit | eBioscience | Cat#: 88-7064-88, RRID:AB_2574990 | |
| Commercial assay or kit | Mouse IL-12 ELISA Kit | MultiSciences Biotech Co., Ltd. | Cat#: 70-EK212/3-96 | |
| Commercial assay or kit | Mouse CXCL1 ELISA Kit | MultiSciences Biotech Co., Ltd. | Cat#: 70-EK296/2-96 | |

*Continued on next page*

*Continued*

| Reagent type (species) or resource | Designation | Source or reference | Identifiers | Additional information |
|---|---|---|---|---|
| Commercial assay or kit | Mouse CXCL2 ELISA Kit | MultiSciences Biotech Co., Ltd. | Cat#: 70-EK2142/2-96 | |
| Commercial assay or kit | Mouse CXCL3 ELISA Kit | Rockland | Cat#: KOA0825 | |
| Commercial assay or kit | mouse CXCL10 ELISA Kit | MultiSciences Biotech Co., Ltd. | Cat#: 70-EK268/2-96 | |
| Antibody | Mouse monoclonal anti-FcγR blocking mAb | BD Biosciences | Clone: 2.4G2, Cat#: 553141, RRID:AB_394656 | FC (1: 100) |
| Antibody | Mouse monoclonal APC/Cy7 conjugated anti-CD45 | BioLegend | Clone: 30-F11, Cat#: 103116, RRID:AB_312981 | FC (1: 100) |
| Antibody | Mouse monoclonal PE conjugated anti-CD11b | BioLegend | Clone: M1/70, Cat#: 101207, RRID:AB_312790 | FC (1: 100) |
| Antibody | Mouse monoclonal PE/Cy7 conjugated anti-Ly6G | BioLegend | Clone: 1A8, Cat#: 127618, RRID:AB_1877261 | FC (1: 100) |
| Antibody | Mouse monoclonal FITC conjugated anti-Ly6C | BD Biosciences | Clone: AL-21, Cat#: 553104, RRID:AB_394628 | FC (1: 100) |
| Antibody | Mouse monoclonal APC conjugated anti-F4/80 | BioLegend | Clone: BM8, Cat#: 123116, RRID:AB_893481 | FC (1: 100) |
| Antibody | Mouse monoclonal APC conjugated anti-CD45 | BioLegend | Clone: 30-F11, Cat#: 103112, RRID:AB_312977 | FC (1: 100) |
| Antibody | Mouse monoclonal FITC conjugated anti-CD3 | BD Biosciences | Clone: 17A2, Cat#: 555274, RRID:AB_395698 | FC (1: 100) |
| Antibody | Mouse monoclonal APC conjugated anti-CD4 | BD Biosciences | Clone: H129.19, Cat#: 553650, RRID:AB_394970 | FC (1: 100) |
| Antibody | Mouse monoclonal PE/Cy7 conjugated anti-CD8 | BioLegend | Clone: 53–6.7, Cat#: 100721, RRID:AB_312760 | FC (1: 100) |
| Antibody | Mouse monoclonal PE conjugated anti-B220 | BD Biosciences | Clone: RA3-6B2, Cat#: 553090, RRID:AB_394620 | FC (1: 100) |
| Antibody | Mouse monoclonal anti-Ly6G antibody | BioXcell | Clone: 1A8, Cat#: BP0075-1, RRID:AB_1107721 | In vivo depletion of neutrophils |
| Antibody | Mouse monoclonal anti-CSF1R | BioXcell | Clone: AFS98, Cat#: BE0213, RRID:AB_2687699 | In vivo depletion of monocytes/macrophages |
| Antibody | Rat monoclonal anti-IgG2a | BioXcell | Clone: 2A3, Cat#: BE0089, RRID:AB_1107769 | Isotype control for anti-mouse Ly6G and anti-mouse CSF1R |
| Antibody | Rabbit monoclonal anti-p38 MAPK | Cell Signaling Technology | Cat#: 9,212 S, RRID: AB_330713 | WB (1: 1000) |
| Antibody | Rabbit monoclonal anti-phospho-p38 MAPK | Cell Signaling Technology | Cat#: 9,211 S, RRID:AB_331641 | WB (1: 1000) |
| Antibody | Rabbit monoclonal anti-ERK MAPK | Cell Signaling Technology | Cat#: 9,102 S, RRID:AB_330744 | WB (1: 1000) |
| Antibody | Mouse monoclonal anti-phospho-ERK MAPK | Cell Signaling Technology | Cat#: 9,106 S, RRID:AB_331768 | WB (1: 1000) |
| Antibody | Rabbit monoclonal anti-JNK MAPK Antibody | Cell Signaling Technology | Cat#: 9,252 S, RRID:AB_2250373 | WB (1: 1000) |

*Continued on next page*

*Continued*

| Reagent type (species) or resource | Designation | Source or reference | Identifiers | Additional information |
|---|---|---|---|---|
| Antibody | Mouse monoclonal anti-phospho-JNK MAPK | Cell Signaling Technology | Cat#: 9,255 S, RRID:AB_2307321 | WB (1: 1000) |
| Antibody | Rabbit monoclonal anti-NF-$\kappa$B p65 | Cell Signaling Technology | Cat#: 8,242 S, RRID:AB_10859369 | WB (1: 1000) |
| Antibody | Rabbit monoclonal anti-phospho-NF-$\kappa$B p65 | Cell Signaling Technology | Cat#: 3,033 S, RRID:AB_331284 | |
| Chemical compound, drug | Thioglycollate medium | Sigma-Aldrich | Cat#: B2551 | |
| Chemical compound, drug | EGTA | Sigma-Aldrich | Cat#: 324,626 | |
| Chemical compound, drug | Zymosan | Sigma-Aldrich | Cat#: Z4250 | |
| Chemical compound, drug | Mueller-Hinton broth | Qingdao Rishui Biotechnologies Co., Ltd | Cat#: 11,816 | |
| Chemical compound, drug | Nutrient Broth | Qingdao Rishui Biotechnologies Co., Ltd | Cat#: 10,204 | |
| Chemical compound, drug | WST-8 | Cayman | Cat#: 18,721 | |
| Chemical compound, drug | Ketamine hydrochloride | R&D Systems | Cat#: 3131/50 | |
| Chemical compound, drug | LPS | Sigma-Aldrich | Cat#: L2630 | |
| Chemical compound, drug | SB202190 | Cell Signaling Technology | Cat#: 8,158 S | |
| Chemical compound, drug | U0126 | Cell Signaling Technology | Cat#: 9,903 S | |
| Chemical compound, drug | SP600125 | Cell Signaling Technology | Cat#: 8,177 S | |
| Chemical compound, drug | BAY11-7082 | Cell Signaling Technology | Cat#: 78,679 S | |
| Chemical compound, drug | LY294002 | Cell Signaling Technology | Cat#: 9,901 S | |

## Cells, bacteria, and peptides

Human monocyte THP-1 cells and rat RBL-2H3 cells were purchased from National Collection of Authenticated Cell Cultures (https://www.cellbank.org.cn/). THP-1 cells were cultured in RPMI 1640 medium supplemented with 10% fetal bovine serum (FBS, Gibco, USA) and antibiotics (100 U/mL penicillin and 100 µg/mL streptomycin). Human THP-1 cell line has been authenticated by STR profiling (*Supplementary file 4*). RBL-2H3 cells were cultured in MEM medium supplemented with NaHCO$_3$ (1.5 g/L), sodium pyruvate (0.11 g/L), 15% FBS (Gibco, USA) and antibiotics (100 U/mL penicillin and 100 µg/mL streptomycin). Peritoneal macrophages and bone marrow-derived neutrophils were isolated from C57BL/6 mice (*Yang et al., 2021*). C57BL/6 mice were intraperitoneally injected with sterile thioglycollate medium (4%, 2 mL). At 4 days post injection, the peritoneal macrophages were collected by flushing with DMEM medium. Mouse bone marrow was rinsed with 5 mL PBS and filtered through a cell strainer (70 micron). After centrifugation at 500 *g* for 5 min, the bone marrow-derived neutrophil pellet was re-suspended in 2 mL PBS. RPMI 1640 diluted Percoll gradient with 72%, 64%, and 54% layers was prepared, and cell suspension was over-layered onto this gradient. Percoll gradient was centrifuged at 950 *g* for 25 min. Neutrophils were collected from the 72%/64% interface, washed with PBS, and centrifuged at 500 *g* for 5 min. Peritoneal macrophages and neutrophils were

confirmed by flow cytometry (*Figure 8—figure supplement 1*, *Figure 10—figure supplement 1*), and were cultured in DMEM and RPMI 1640 medium, respectively, supplemented with 10% FBS (Gibco, USA) and antibiotics (100 U/mL penicillin and 100 µg/mL streptomycin). Cells were maintained under an atmosphere of 5% $CO_2$ at 37°C. Fluorescent quantitative PCR (qPCR, forward primer, 5'-GGGA GCAAACAGGATTAGATACCCT-3', reverse primer, 5'-TGCACCATCTGTCACTCTGTTAACCTC-3') was performed to confirm that the cell lines were negative for mycoplasma contamination.

Gram-positive bacteria, Gram-negative bacteria, and fungi were cultured at 37°C in Luria-Bertani (LB) broth. Aquatic pathogenic bacteria were cultured at 25°C in nutrient broth.

Synthetic peptides were purchased from Synpeptide Co. Ltd (Shanghai, China). The crude peptide was purified by reversed-phase high performance liquid chromatography (RP-HPLC) and analysed by mass spectrometry to confirm the purity higher than 98%.

## Experiment animals

Both adult healthy tree frogs of *P. puerensis* (21–30 g) were captured from Pu'er, Yunnan Province, China (24.786°N, 101.362°E). *P. puerensis* was not endangered or protected species, and no specific permissions were required for the sampling location/activity. Tree frogs were randomly housed in freshwater tanks in a recirculating system with filtered water, fed with mealworm larvae *Tenebrio molitor* and refreshed with water once a day. C57BL/6 mice (female, 18–20 g) were purchased from Shanghai Slac Animal Inc, and *Rag1*$^{-/-}$ mice (female, 18–20 g) were purchased from Model Animal Research Center of Nanjing University. Mice were housed in pathogen-free facility. Animal experiments were performed in accordance with the Institutional Animal Care and Use Committee of Soochow University, and all research protocols were approved by the Animal Ethical Committee of Soochow University. All surgery of animals was performed under pentobarbital sodium anaesthesia with minimum fear, anxiety, and pain.

## Mature peptide isolation

Skin secretions were collected according to previous study (*Li et al., 2007*). Briefly, frogs were stimulated by anhydrous ether, and a total of about 500 mL skin secretions in PBS were quickly collected, centrifuged, and lyophilised. Lyophilised *P. puerensis* skin secretion was dissolved in phosphate-buffered saline (PBS, 0.1 M, pH 6.0) and separated by molecular sieving fast protein liquid chromatography (FPLC) on GE ÄKTA pure system using a Superdex 75 10/300 GL column (10 × 300 mm, 24 mL volume, GE, USA). Fractions were pooled and further purified by RP-HPLC on a C18 column (25 × 0.46 cm, Waters, USA) for two times. The eluted peaks from RP-HPLC were collected for purity assay using matrix-assisted laser desorption ionisation time-of-flight mass spectrometry (MALDI-TOF MS) on an UltraFlex I mass spectrometer (Bruker Daltonics, Germany). The amino acid sequence of the purified peptide was obtained by automated Edman degradation analysis on an Applied Biosystems-pulsed liquid-phase sequencer (model ABI 491, USA).

## cDNA cloning

Skin total RNA extraction, mRNA isolation and cDNA library construction were performed according to previous methods (*Wei et al., 2015*). About $5.6 × 10^5$ independent colonies were produced in the cDNA library. Two primers, an antisense primer, 5'-TTGTCTGCCTCCTCGGCTTCC-3', designed according to the conserved domain of amphibian cathelicidins, and the 5' PCR primer, 5'-AAGCAGTG GTATCAACGCAGAGT-3' supplied by cDNA library construction kit, were used to clone the 5' fragment that encoding the precursor of *Popu*CATH. The full length cDNA encoding the precursor of *Popu*CATH was obtained by a sense primer, 5'-ATGGCGCTCGCTGCTGCACTC-3' designed according to the 5' fragment of *Popu*CATH precursor, and 3' PCR primer, 5'-ATTCTAGAGGCCGAGGCGGCCG-3' provided by the kit. PCR procedure for cDNA cloning was 95 °C for 5 min, and 30 cycles of 95 °C for 30 s, 56 °C for 30 s, 72 °C for 1 min, followed by an extension step at 72 °C for 8 min.

## Toxic side effects to mammalian cells and mice

For cytotoxicity assay, mouse peritoneal macrophages or THP-1 cells were seeded into 96-well plates ($5 × 10^5$ cells/well, 200 µL). *Popu*CATH (25, 50, 100, and 200 µg/mL) was added to each well. After culture for 24 h, 10 µL of CCK-8 reagent was added to each well. The absorbance at 450 nm was recorded on a microplate reader after incubation for 1 h (*Yang et al., 2021*).

For hemolysis assay, mouse erythrocytes and rabbit erythrocytes were washed with 0.9% saline and incubated with a series of two-fold dilutions of *PopuCATH* (25, 50, 100, and 200 µg/mL) at 37 °C. After incubation for 30 min, the erythrocytes were centrifuged at 1,000 *g* for 5 min and monitored at 540 nm. Triton X-100 (1%) treatment was determined as 100% hemolysis. Hemolytic activity was expressed as the percentage of the Triton X-100-treated group (*Wei et al., 2013*).

For immunogenicity assay, mesenteric lymph nodes (MLN) and spleen were collected and filtered through at 70 µm cell strainer (Falcon, Corning, USA). After erythrocytes were lysed with ACK Lysis Buffer (Solarbio, Beijing, China) for 5 min, cells were suspended in RPMI 1640 (2%FBS), and added to 96-well plates ($5 \times 10^4$ cells/well, 200 µL). A final concentration of 25, 50, 100, or 200 µg/mL of *PopuCATH*, or 2 µg/mL of concanavalin A (Con A, Sigma-Aldrich, Shanghai, China) was added and incubated at 37°C for 24 h. CCK-8 reagent (10 µL, Dojindo, Shanghai, China) was added. After incubation at 37°C for 1 h, the absorbance at 450 nm was measured on a microplate reader (*Mendez et al., 2005*).

For hypersensitivity assay, RBL-2H3 cells were seeded in 96-well plates ($2 \times 10^4$ cells/well, 200 µL) and cultured overnight. A final concentration of 25, 50, 100, or 200 µg/mL of *PopuCATH* or human catheticidin LL-37 (positive control) was added and incubated at 37°C for 0.5 hr. The supernatant was collected and incubated with 4-nitrophenyl-N-acetyl-B-D-glucosaminide substrate at 37°C for 1 hr. The absorbance at 405 nm was measured on a microplate reader (*Scott et al., 2007*).

For complement assay, mouse serum was treated with PBS, EGTA inhibitor (10 mM, Sigma-Aldrich, Shanghai, China), zymosan (0.5 mg/mL, Sigma-Aldrich, Shanghai, China), *PopuCATH* (25, 50, 100, and 200 µg/mL) at 37°C for 1 hr. C3a des-Arg was measured by ELISA (Wuhan Fine Biotech, China) (*Scott et al., 2007*).

For in vivo acute toxicity assay, C57BL/6 (female, 18–20 g, n = 6) were intraperitoneally injected with *PopuCATH* at dose of 10, 20, and 40 mg/kg, respectively. At 24 hr post injection, kidneys, livers, hearts and spleens were collected for H&E staining. The alanine aminotransferase (ALT) and creatinine in the serum were measured by ALT assay kit (Nanjing Jiancheng Bioengineering Institute, China) and the creatinine assay kit (Nanjing Jiancheng Bioengineering Institute, China), respectively (*Yu et al., 2017*).

## In vitro antimicrobial assay

A standard two-fold broth microdilution method was used to evaluate the MIC of *PopuCATH* against microbes. Gram-positive bacteria, Gram-negative bacteria, and fungi were diluted with Mueller-Hinton broth, and aquatic pathogenic bacteria were diluted with nutrient broth to $10^5$ CFU/mL. Series of two-fold *PopuCATH* dilutions were prepared in 96-well plates (50 µL/well). An equal volume of microbial dilution was added and cultured at 37°C (for Gram-positive bacteria, Gram-negative bacteria, and fungi) or 25°C (for aquatic pathogenic bacteria) for 18 hr. Catheticidin-PY from *P. yunnanensis* served as positive control. The minimal concentrations at which no visible growth of microbes occurred were defined as MIC values (*Wei et al., 2013*).

Bacterial killing kinetics were examined as described previously (*Wei et al., 2013*). Microbes in exponential phase were diluted in Mueller-Hinton broth (*E. coli* ATCC25922, *S. aureus* ATCC25923, and *C. albicans* ATCC2002) or nutrient broth (*A. hydrophila*) at density of $10^5$ CFU/mL. *PopuCATH* (200 µg/mL), catheticidin-PY (PY, 1× MIC) or an equal volume of PBS (solvent of peptide) was incubated with microbial dilution at 37°C or 25°C for 0, 1, 2, 3, and 4 hr, respectively. At each time point, mixture of peptide and microbe was diluted in Mueller-Hinton or nutrient broth for 1000 folds, and microbial dilution (50 µL) was coated on Mueller-Hinton or nutrient broth agar plates. Microbial colonies were counted after culture at 37°C or 25°C for 12 hr.

Microbial metabolic activities were assayed according previous method (*Scott et al., 2007*). *E. coli* ATCC25922, *S. aureus* ATCC25923, *C. albicans* ATCC2002, and *A. hydrophila* in exponential phase were diluted in DMEM at density of $10^5$ CFU/mL, and *PopuCATH* (200 µg/mL), catheticidin-PY (PY, 1× MIC) or PBS (solvent of peptide) was added. Microbial dilution (100 µL/well) was added to 96-well plates. After the addition of WST-8 (10 µL/well, Cayman, Ann Arbor, USA), the plates were incubated at 37°C or 25°C for 1, 2, 3, and 4 hr, and absorbance was monitored at 255 nm. Metabolic activity was expressed as the percentage of the PBS-treated group.

Scanning electron microscope (SEM) assay was used to examine if *PopuCATH* impairs the bacterial surface morphology. *E. coli* ATCC25922 and *S. aureus* ATCC25923 were cultured in Mueller-Hinton

broth to exponential phase, washed and diluted using PBS ($10^5$ CFU/mL). *Popu*CATH (200 µg/mL), cathelicidin-PY (PY, 1× MIC) or PBS was added into the bacterial dilution and incubated at 37°C. After incubation for 30 min, bacteria were centrifuged (1000 *g* for 10 min) and fixed for SEM assay according to standard operating protocols. The bacterial surface morphology was observed using a Hitachi SU8010 SEM (*Wei et al., 2013*).

## In vivo antimicrobial assay

In tree frogs (n = 5, 21–30 g), *Popu*CATH (10 mg/kg) was intraperitoneally injected at 8 or 4 hr prior to (–8 or –4 hr), or 4 hr after ( + 4) *S. aureus* ATCC25923 inoculation ($10^8$ CFU/frog, intraperitoneal injection). At 18 hr post bacterial challenge, peritoneal lavage was collected for bacterial load assay.

In C57BL/6 mice (female, 18–20 g, n = 6), *Popu*CATH (10 mg/kg) was intraperitoneally injected at eight or 4 hr prior to (–8 or –4 hr), or 4 hr after ( + 4) Gram-negative (*E. coli, A. baumannii*) or Gram-positive (*S. aureus* or methicillin-resistant *S. aureus*, MRSA) bacterial inoculation (2 × $10^7$ CFUs/mouse, intraperitoneal injection). At 18 hr post bacterial inoculation, peritoneal lavage was collected for bacterial load assay, serum was collected for cytokine assay, and lungs were taken for histopathological assay (*Yang et al., 2021*).

In order to further investigate its prophylactic efficacy against bacterial infection, *Popu*CATH (10 mg/kg) was given through intravenous or intramuscular injection at 4 hr prior to *E. coli* inoculation (2 × $10^7$ CFUs/mouse, intraperitoneal injection). At 18 hr post bacterial inoculation, peritoneal lavage was collected for bacterial load assay.

The protective efficacy of *Popu*CATH was also evaluated in septic mice induced by a lethal bacterial inoculation or CLP. For lethal bacterial challenge, C57BL/6 mice (female, 18–20 g, n = 6) were intraperitoneally injected with *Popu*CATH (10 mg/kg) 4 hr prior to *E. coli* (4 × $10^7$ CFUs/mouse, intraperitoneal injection) or MRSA (6 × $10^8$ CFUs/mouse, intraperitoneal injection) inoculation. The survival rates of mice were monitored for 7 days (*Yang et al., 2021*). To compare the protective efficacy of *Popu*CATH with other peptides, the protective efficacy of LL-37 and IDR-1 were simultaneously evaluated at the same condition. For CLP-induced sepsis, C57BL/6 mice (female, 18–20 g, n = 6) were intraperitoneally injected with *Popu*CATH (10 mg/kg) at 8 and 4 hr (two times) prior to CLP. At 4 hr post the last injection of *Popu*CATH, mice were anaesthesied with ketamine hydrochloride (100 mg/kg), and the abdominal cavity of mice was opened in layers. The cecum was ligated 1.0 cm from the end, a through-and-through puncture was operated using an 18-gauge needle. A small droplet of faeces was extruded for ensuring the patency of the puncture site. Then, the cecum was returned back to the abdominal cavity. A laparotomy but no CLP mice served as control. After CLP, the survival rates of mice were monitored for 7 days (*Yang et al., 2021*).

## In vivo chemotaxis assay

C57BL/6 mice (female, 18–20 g, n = 6) were intraperitoneally injected with *Popu*CATH (10 mg/kg) dissolved in 0.2 mL PBS. The same volumes of PBS and LPS (20 µg/mouse, from *E. coli* O111:B4, Sigma-Aldrich, Shanghai, China) were used as negative control and positive control, respectively. Cells in peritoneal lavage and peripheral blood were collected at 4 and 8 hr post injection, respectively. For chemotactic kinetics assay, cells in peritoneal lavage and peripheral blood were collected at 4, 8, 24, 48, 72 hr post injection, respectively. Collected cells were incubated with anti-FcγR blocking mAb (clone 2.4G2). After incubation at 4°C for 30 min, cells were washed with PBS and re-suspended in PBS. For myeloid cell analysis, cells were stained with APC-Cy7/anti-CD45 (clone 30-F11), PE/anti-CD11b (clone M1/70), PE/Cy7/anti-Ly-6G (clone 1A8), FITC/anti-Ly6C (clone AL-21), and APC/F4/80 (clone BM8) at 4°C for 30 min. For lymphoid cell analysis, cells were stained with APC/anti-CD45 (clone 30-F11), FITC/anti-CD3 (clone 17A2), ACP-Cy7/anti-CD4 (clone H129.19), PE/Cy7/ anti-CD8 (clone 53–6.7), and PE/B220 (clone RA3-6B2) at 4°C for 30 min. The stained cells were washed and analysed by a flow cytometer FACS Canto II (BD Biosciences) with FlowJo seven software (Tree Star) (*Scott et al., 2007*; *Yang et al., 2021*).

*P. puerensis* (21–30 g, n = 5) were intraperitoneally injected with *Popu*CATH (10 mg/kg, dissolved in 0.2 mL PBS) or PBS. At 4 and 8 hr post injection, total cells in peritoneal lavage were counted using a hemocytometer, and the cells were observed under an optical microscope after Wright-Giemsa staining.

## Protective efficacy of *Popu*CATH in myeloid or lymphoid cell-deficient mice

Neutrophils and monocytes/macrophages were depleted by intraperitoneal injection of anti-Ly6G antibody and anti-CSF1R antibody, respectively (*Yang et al., 2021*). Anti-Ly6G antibody (500 µg/mouse) or anti-rat IgG2a isotype antibody was injected into C57BL/6 mice (female, 18–20 g, n = 6) on day 0, and day 2, respectively. Anti-CSF1R antibody or anti-rat IgG2a isotype antibody was injected into C57BL/6 mice (female, 18–20 g, n = 6) at doses of 1 mg per mouse on day 0 followed by 0.3 mg per mouse on day 1 and day 2, respectively.

*Popu*CATH (10 mg/kg) was intraperitoneally injected into neutrophil-depleted mice (on day 3), monocyte/macrophage-depleted mice (on day 3), and *Rag1*$^{-/-}$ mice (female, 18–20 g, n = 6) 4 hr prior to *E. coli* or *S. aureus* inoculation ($2 \times 10^7$ CFUs/mouse, intraperitoneal injection). At 18 hr post bacterial challenge, the peritoneal lavage was collected for bacterial load assay (*Yang et al., 2021*).

## In vitro chemotaxis assay

Neutrophils or macrophages were suspended in RPMI 1640 ($5 \times 10^6$ cells/mL, 2% FBS), and 100 µL of cell suspension was added to the 3.0 µm (for neutrophils) or 5.0 µm (for macrophages) pore-size Transwell filters (the upper chamber) in a 24-well format. A total of 500 µL of *Popu*CATH (10 µM, dissolved in 2% FBS RPMI 1640 medium) or medium was added to the lower chamber. After culture at 37°C for 8 hr, neutrophils or macrophages in the lower chamber were counted using a hemocytometer. The increased cells in the lower chamber were determined as the migrated cells (*Yang et al., 2021*).

For the co-cultured system, 500 µL of macrophage suspension in RPMI 1640 ($5 \times 10^6$ cells/mL, 2% FBS) were added to the lower chamber, and 100 µL of neutrophil or macrophage suspension in RPMI 1640 ($5 \times 10^6$ cells/mL, 2% FBS) was added to the 3.0 µm (for neutrophils) or 5.0 µm (for macrophages) pore-size Transwell filters (the upper chamber). A total of 500 µL of *Popu*CATH (10 µM, dissolved in medium) or medium was added to the lower chamber. After culture at 37°C for 8 hr, neutrophils or macrophages in the upper chamber were counted using a hemocytometer. The reduced cells in the upper chamber were determined as the migrated cells (*Yang et al., 2021*).

## Regulatory effects of *Popu*CATH on macrophages

For chemokine/cytokine production assay, macrophages were seeded in 24-well plates ($5 \times 10^5$ cells/well, 2%FBS DMEM) and cultured with *Popu*CATH (5, 10, and 20 µM). After incubation at 37°C for 4 hr, the cells were collected for total RNA extraction using Trizol reagent. SYBR green qPCR master mix was used to for two-step qPCR assay after cDNA synthesis using PrimeScript RT reagent kit. Gene expression was normalised to the expression level of *Actb*. Primers for qPCR were listed in **Supplementary file 5**. The supernatant was collected for chemokine/cytokine production determination using ELISA kits (*Yang et al., 2021*).

For signaling pathway assay, macrophages seeded in 24-well plates ($5 \times 10^5$ cells/well, 2%FBS DMEM) were pre-incubated with inhibitor of p38 (SB202190, 10 µM), ERK (U0126, 10 µM), JNK (SP600125, 10 µM), NF-κB (BAY11-7082, 2 µM), or PI3K (LY294002, 10 µM) for 1 hr, respectively, and *Popu*CATH (10 µM) was added and incubated for 4 hr. Chemokine production in the supernatant were quantified by ELISA. Next, macrophages were plated to six-well plates ($2 \times 10^6$/well, 2%FBS DMEM) and cultured with *Popu*CATH (10 µM) for 15, 30, and 45 min, respectively. LPS (100 ng/mL, incubation for 30 min) served as positive control. Macrophages were lysed for detecting the protein levels of p38, phospho-p38, ERK, phospho-ERK, JNK, phospho-JNK, p65, and phospho-p65 by western blot analysis (*Yang et al., 2021*).

For macrophage phagocytosis assay, macrophages were pre-incubated with *Popu*CATH (10 µM) for 2 hr, *S. aureus* and *E. coli* bacteria were preloaded with 10 µM CFSE fluorescent dye (Molecular Probes, Invitrogen) in PBS for 30 min at 37°C. Bacteria were killed by incubation with 1% paraformaldehyde in PBS for 1 hr at 37°C, and washed five times in PBS. Macrophages were incubated with the CFSE-loaded bacterial particles at multiplicity of infection 100. After incubation for indicated time points, the cells were washed, the extracellular fluorescence were quenched with trypan blue (15 mg/mL) in PBS, and the CFSE fluorescence were analysed by flow cytometry as a measure of the phagocytic uptake of the bacterial particles (*Nijnik et al., 2010*).

## Regulatory effects of *Popu*CATH on neutrophils

The effects of *Popu*CATH (10 µM) on neutrophil phagocytosis was assayed similar with macrophages phagocytosis as mentioned above. Neutrophils were pre-incubated with *Popu*CATH (10 µM) for 2 hr, and were incubated with CSFE-loaded *S. aureus* and *E. coli* particles at multiplicity of infection 100. At indicated time points, CFSE fluorescence were analysed by flow cytometry (*Nijnik et al., 2010*).

For neutrophil extracellular traps (NETs) assay, neutrophils suspended in 2% FBS RPMI 1640 was seeded in an eight well-cover slip chamber (200 µL/well, $1 \times 10^6$ cells/mL). *Popu*CATH (10 µM), PBS, or PMA (100 nM) was added and incubation at 37°C for 4 hr. Neutrophils were stained with DAPI (Invitrogen, USA) or anti-Cit-H3 (Abcam, USA), respectively. Nuclei and NETs were observed under a confocal microscope ( × 60, Nikon, Japan) (*He et al., 2019*).

## Statistical analysis

Data were represented as mean ± SD. Statistical significance was determined by an unpaired two-tailed Student's *t* tests for two-group comparison, and was determined by ANOVA followed by Bonferroni post hoc analysis for multiple-group comparison. All statistical analysis was performed using GraphPad Prism software version 5.0. A p value less than 0.05 was considered as statistically significant.

## Additional information

### Funding

| Funder | Grant reference number | Author |
|---|---|---|
| National Natural Science Foundation of China | 31870868 | Lin Wei |
| National Natural Science Foundation of China | 31970418 | Hailong Yang |
| National Natural Science Foundation of China | 81373380 | Hailong Yang |
| National Natural Science Foundation of China | 81802023 | Jing Wu |
| Key Research & Development Plan in Social Development of Jiangsu Province | BE2020652 | Lin Wei |
| Priority Academic Program Development of Jiangsu Higher Education Institutions | | Lin Wei |

The funders had no role in study design, data collection and interpretation, or the decision to submit the work for publication.

### Author contributions

Yang Yang, Data curation, Investigation, Methodology, Project administration, Writing - original draft; Jing Wu, Data curation, Funding acquisition, Investigation, Writing - original draft; Qiao Li, Data curation, Methodology, Writing - original draft; Jing Wang, Data curation, Investigation, Methodology; Lixian Mu, Methodology, Resources; Li Hui, Methodology, Resources, Supervision, Writing – review and editing; Min Li, Conceptualization, Funding acquisition, Methodology, Project administration, Supervision, Writing – review and editing; Wei Xu, Hailong Yang, Conceptualization, Funding acquisition, Project administration, Resources, Supervision, Writing – review and editing; Lin Wei, Conceptualization, Data curation, Funding acquisition, Investigation, Project administration, Resources, Supervision, Writing – review and editing

### Author ORCIDs

Lin Wei (iD) http://orcid.org/0000-0003-3359-2471

## Ethics

Animal experiments were performed in accordance with the Institutional Animal Care and Use Committee of Soochow University, and all research protocols were approved by the Animal Ethical Committee of Soochow University. All surgery of mice was performed under pentobarbital sodium anesthesia with minimum fear, anxiety and pain.

## Decision letter and Author response

Decision letter https://doi.org/10.7554/eLife.72849.sa1
Author response https://doi.org/10.7554/eLife.72849.sa2

## Additional files

### Supplementary files

• Supplementary file 1. Physico-chemical parameters of *Popu*CATH.

• Supplementary file 2. Secondary structural components of *Popu*CATH in aqueous solution and membrane-mimetic solution.

• Supplementary file 3. MIC values of *Popu*CATH against Gram-negative bacteria, Gram-positive bacteria, fungi, and aquatic bacteria.

• Supplementary file 4. THP-1 cell line authentication report by STR profiling.

• Supplementary file 5. Primers for qPCR.

• Transparent reporting form

### Data availability

Sequencing data have been deposited in GenBank (accession number: KY391886). All data generated or analysed during this study are included in the manuscript, supporting files, and source data.

The following dataset was generated:

| Author(s) | Year | Dataset title | Dataset URL | Database and Identifier |
|---|---|---|---|---|
| Wei L, Yang H | 2016 | Rhacophorus puerensis cathelicidin precursor, mRNA, complete cds | https://www.ncbi.nlm.nih.gov/nuccore/KY391886 | NCBI GenBank, KY391886 |

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
