## [Editor Report]

This manuscript describes for the first time a novel cathelicidin, namely, *Popu*CATH which is able to prevent the development of infection by different bacterial species in two different animal models, frog and mouse. The mechanism of action is exerted through priming of neutrophil efflux.

---

## [Decision Letter]

**Decision letter after peer review:**

Thank you for submitting your work entitled "A non-bactericidal cathelicidin provides prophylactic efficacy against bacterial infection by driving phagocyte influx" for further consideration by *eLife*. Your article has been reviewed by 2 peer reviewers, including Evangelos Giamarellos-Bourboulis as the Reviewing Editor and Reviewer #1, and the evaluation has been overseen by Carla Rothlin as Senior Editor. The following individual involved in the review of your submission have agreed to reveal their identity: Xueqing Xu (Reviewer #2).

The manuscript has been improved but there are some remaining issues that need to be addressed, as outlined below:

The study by Yang et al., identified a novel cathelicidin antimicrobial peptide named PopuCATH from the tree frog, Polypedates puerensis. in vitro antimicrobial assay showed that PopuCATH didn't show any direct anti-bacterial activity. in vivo antimicrobial assay showed that intraperitoneal injection of PopuCATH before bacterial inoculation significantly reduced the bacterial load in tree frogs and mice, and reduced the inflammatory response induced by bacterial inoculation in mice. PopuCATH pretreatment also increased the survival rates of septic mice induced by a lethal dose of bacterial inoculation or CLP. The underlying mechanism revealed that intraperitoneal injection of PopuCATH significantly drove the leukocyte influx in both frogs and mice. In mice, PopuCATH rapidly drove neutrophil, monocyte/macrophage influx in mouse abdominal cavity and peripheral blood with a negligible impact on T and B lymphocytes, and neutrophils, monocytes/macrophages, but not T and B lymphocytes, were required for the preventive efficacy of PopuCATH. in vitro cell migration assay indicated that PopuCATH did not directly act as chemoattractant for phagocytes, but PopuCATH obviously drove phagocyte migration when it was cultured with macrophages. Further investigation showed that PopuCATH significantly elicited chemokine/cytokine production in macrophages through activating p38/ERK MAPKs and NF-κB p65. Incubation of PopuCATH with neutrophils also markedly enhanced neutrophil phagocytosis via promoting the release of NETs. These findings revealed the anti-bacterial action and relative mechanism of a non-bactericidal cathelicidin against bacterial infection, which are novel and interesting. The main strength of this submission is the description of an entire novel peptide. The main limitation is the lack of sufficient explanation why popuCATH is able to prevent the development of an infection but has no effect at all when injected in parallel or after whole bacteria.

Suggestions for improvement:

– The authors need to run time-kill assays where in the interaction of popuCATH with antimicrobial is tested. This will provide some evidence if there is a future for this peptide in therapeutics.

– The authors cannot discuss on the anti-inflammatory activity of popuCATH since the attenuation of the inflammation may be secondary to the decrease of bacterial growth.

– It is very difficult to explain the total absence of effect when popuCATH is given after induction of infection or in parallel to bacterial challenge. This diversity generates ambiguity on the real physiological meaning of this peptide.

– Does popuCHATH has more than one gene per chromosome like defensins?

– In this study, a total of 16 amino acids at the N-terminus of the purified peptide were determined as SRGGRGGRGGGGSRGG by automated Edman degradation. The authors should have designed an antisense primer according to this partial mature peptide. However, the authors designed an antisense primer (5'-TTGTCTGCCTCCTCGGCTTCC-3') according to the conserved domain of amphibian cathelicidins. Why?

– In Figures 2A, 2B, 2C, 3A, 3C, 3D, 3E, and Supplementary file 3, the highest concentration of PopuCATH is 200 µg/mL. Why did the authors select 200 µg/mL, other than 300, 400, 500…µg/mL as the highest concentration?

– In both frogs and mice, PopuCATH (10 mg/kg) treatment at 8 h or 4 h before bacterial inoculation significantly reduced the bacterial load in tree frogs, but PopuCATH treatment at 4 h after bacterial inoculation did not significantly reduce the bacterial load. The results indicated that PopuCATH exhibited prophylactic efficacy, but not therapeutic efficacy, against bacterial infection. I would like to known the possible reasons why PopuCATH has prophylactic efficacy, but not therapeutic efficacy. At least, the authors should discuss the data.

– In Figure 8, PopuCATH-mediated chemokine/cytokine production in macrophages and mice is somewhat different. What is(are) the possible reason(s)? I suggest the authors discuss this difference.

– A previous study showed that a frog-derived glycine-rich cathelicidin exhibited direct antimicrobial activity (Hao et al., 2012). In this study, PopuCATH is also a glycine-rich cathelicidin. But PopuCATH did not show any direct antimicrobial activity at concentration up to 200 µg/mL. Why? Please discuss this phenomenon.

– Line 103-104, the authors described that a novel naturally occurring cathelicidin was identified from the skin of tree frog, P. puerensis. Line 497, the authors described that synthetic peptides were purchased from Synpeptide Co. Ltd. That is to say, some results were obtained by using naturally occurring PopuCATH, and the others were obtained by using synthetic PopuCATH. Please describe which results were obtained by using naturally occurring/synthetic PopuCATH in the responding Materials and methods section.

– In Figure 8, the mRNA levels of chemokines/cytokines in macrophages induced by PopuCATH were tested by Q-PCR. Please list the primers for Q-PCR assay.

– To assay the effector cells, the protective efficacy of PopuCATH was evaluated in neutrophil or monocyte/macrophage depletion mice by intraperitoneal injection of anti-Ly6G antibody (clone 1A8, BioXcell) and anti-CSF1R antibody (clone AFS98, BioXCell), respectively. However, I the scavenging efficiency of intraperitoneal injection of anti-Ly6G antibody (clone 1A8, BioXcell) and anti-CSF1R antibody (clone AFS98, BioXCell) should be provided, respectively.

---

## [Author Response]

Suggestions for improvement:– The authors need to run time-kill assays where in the interaction of popuCATH with antimicrobial is tested. This will provide some evidence if there is a future for this peptide in therapeutics.

We tested the interaction of *Popu*CATH (200 µg/mL) with ampicillin (0.5×MIC, 1×MIC, 2×MIC) against *E. coli* ATCC25922 by time-kill assay. We found that the presence of *Popu*CATH (200 µg/mL) did not promote the clearance of *E. coli* by 0.5×MIC, 1×MIC, 2×MIC of ampicillin, respectively. Therefore, it is not a good evidence for usage of *Popu*CATH and antimicrobial in combination, and we did not show this data as well as discuss this data in the revised manuscript.

– The authors cannot discuss on the anti-inflammatory activity of popuCATH since the attenuation of the inflammation may be secondary to the decrease of bacterial growth.

We have revised it in the manuscript as follows:

“In mouse model, the production of chemokines/cytokines in mouse abdominal cavity peaked at 4 h post injection of PopuCATH. The dynamic of CXCL1, CXCL2, and CXCL3 production is consistent with the dynamic of neutrophil, monocyte/macrophage recruitment in the mouse abdominal cavity and peripheral blood. Although the pretreatment with *Popu*CATH significantly induced the production of chemokines (CXCL1, CXCL2, and CXCL3) as well as pro-inflammatory cytokines (IL-1β and IL-6), *Popu*CATH ultimately attenuated the inflammatory response by decrease of TNF-α, IL-1β and IL-6 levels post Gram-negative and Gram-positive bacterial infection. Cathelicidins are able to block Toll like receptor (TLR)-mediated inflammatory responses, including those mediated by TLR2 and TLR4 (Coorens et al., 2017; Mookherjee et al., 2006; Wei et al., 2013). In this study, *Popu*CATH didn’t affect LTA- and LPS-stimulated inflammatory responses in mouse peritoneal macrophages (*Figure 8—figure supplement 3*), suggesting that the anti-inflammatory effects of *Popu*CATH are independent of TLRs, and the attenuation of the inflammation may be secondary to the decrease of bacterial growth.”

– It is very difficult to explain the total absence of effect when popuCATH is given after induction of infection or in parallel to bacterial challenge. This diversity generates ambiguity on the real physiological meaning of this peptide.

At the dose of 10 mg/kg, *Popu*CATH did not exhibit therapeutic efficacy against bacterial infection. In order to evaluate whether it has therapeutic efficacy at high doses, we increased the dose of *Popu*CATH up to 20 and 40 mg/kg. We found that 20 mg/kg and 40 mg/kg of *Popu*CATH significantly reduced the bacterial load when it was given at 4 h after *E. coli* inoculation (*Figure 4—figure supplement 2*). It is possibly that bacteria have colonized in mice at 4 h after bacterial inoculation, which need a higher dose of *Popu*CATH to drive more phagocytes for bacterial clearance. And we have added this finding in the Discussion section.

– Does popuCHATH has more than one gene per chromosome like defensins?

As described in the second paragraph of the Introduction section, we have identified a cathelicidin antimicrobial peptide from the skin of *Polypedates puerensis* previously (PMID: 28593346), implying that *Popu*CHATH probably has more than one gene per chromosome.

– In this study, a total of 16 amino acids at the N-terminus of the purified peptide were determined as SRGGRGGRGGGGSRGG by automated Edman degradation. The authors should have designed an antisense primer according to this partial mature peptide. However, the authors designed an antisense primer (5'-TTGTCTGCCTCCTCGGCTTCC-3') according to the conserved domain of amphibian cathelicidins. Why?

We have designed series of degenerate primers based on the amino acid sequence at the N-terminus. But we failed to clone the cDNA sequence. It is possibly that the amino acid sequence at the N-terminus of *Popu*CHATH is a glycine-rich peptide fragment, and the degenerate primers based on this fragment have high degeneracy. Therefore, we then designed an antisense primer (5'-TTGTCTGCCTCCTCGGCTTCC-3') according to the conserved domain of amphibian cathelicidins.

– In Figures 2A, 2B, 2C, 3A, 3C, 3D, 3E, and Supplementary file 3, the highest concentration of PopuCATH is 200 µg/mL. Why did the authors select 200 µg/mL, other than 300, 400, 500…µg/mL as the highest concentration?

According to a previous study (PMID: 17384586), we selected 200 µg/mL as the highest concentration for antimicrobial assay in vitro. Accordingly, if a peptide (drug) did not exhibit antimicrobial activity at a concentration up to 200 µg/mL in vitro, we usually think that it lacks direct antimicrobial activity. Based on the highest concentration used in antimicrobial assay in vitro, we also selected 200 µg/mL as the highest concentration for side effects assay in vitro.

– In both frogs and mice, PopuCATH (10 mg/kg) treatment at 8 h or 4 h before bacterial inoculation significantly reduced the bacterial load in tree frogs, but PopuCATH treatment at 4 h after bacterial inoculation did not significantly reduce the bacterial load. The results indicated that PopuCATH exhibited prophylactic efficacy, but not therapeutic efficacy, against bacterial infection. I would like to known the possible reasons why PopuCATH has prophylactic efficacy, but not therapeutic efficacy. At least, the authors should discuss the data.

In the last version of our manuscript, *Popu*CATH did not exhibit therapeutic efficacy against bacterial infection at dose of 10 mg/kg. In order to evaluate whether it has therapeutic efficacy at high doses, we increased the dose of *Popu*CATH in the revised manuscript. When the dose of *Popu*CATH was increased to 20 mg/kg, *Popu*CATH significantly reduced the bacterial load when it was given at 4 h after bacterial inoculation (Figure 4—figure supplement 2).

– In Figure 8, PopuCATH-mediated chemokine/cytokine production in macrophages and mice is somewhat different. What is(are) the possible reason(s)? I suggest the authors discuss this difference.

The expression profiles of chemokines/cytokines in vitro and in vivo are somewhat different, but we observed that the profiles of the major chemokines/cytokines induced by *Popu*CATH, such as CXCL1, CXCL2, and CXCL3, are similar. It is possibly that *Popu*CATH-induced chemokines/cytokines in vivo are consumed timely. In addition, macrophage is the unique effector cell type of *Popu*CATH in vitro. While there are many other cell types in vivo, such as monocytes and neutrophils, and we cannot exclude these cells are responsive to *Popu*CATH and subsequently produce chemokines/cytokines. These points may explain the subtle differences of *Popu*CATH-mediated chemokine/cytokine production in macrophages and mice. And we have discussed it in the Discussion section.

– A previous study showed that a frog-derived glycine-rich cathelicidin exhibited direct antimicrobial activity (Hao et al., 2012). In this study, PopuCATH is also a glycine-rich cathelicidin. But PopuCATH did not show any direct antimicrobial activity at concentration up to 200 µg/mL. Why? Please discuss this phenomenon.

We have discussed it in the Discussion section as follows:

“The first frog-derived cathelicidin is also rich in glycine residues. But it has different amino acid sequence with *Popu*CATH and exhibits direct anti-bacterial activity unlike *Popu*CATH (Hao et al., 2012). Cathelicidin antimicrobial peptides display a high structural diversity, and the diverse structures are responsible for their diverse functions. Accordingly, it is not difficult to understand that these two frog cathelicidins have different functions against bacteria.”

– Line 103-104, the authors described that a novel naturally occurring cathelicidin was identified from the skin of tree frog, P. puerensis. Line 497, the authors described that synthetic peptides were purchased from Synpeptide Co. Ltd. That is to say, some results were obtained by using naturally occurring PopuCATH, and the others were obtained by using synthetic PopuCATH. Please describe which results were obtained by using naturally occurring/synthetic PopuCATH in the responding Materials and methods section.

Naturally occurring *Popu*CATH was just used in mass spectrometry and Edman degradation assay. Synthetic *Popu*CATH was used in other experiments. And we have described this point in the Materials and methods section.

– In Figure 8, the mRNA levels of chemokines/cytokines in macrophages induced by PopuCATH were tested by Q-PCR. Please list the primers for Q-PCR assay.

We are sorry to lose the primers for Q-PCR assay, and we have listed the primers in Supplementary file 5.

– To assay the effector cells, the protective efficacy of PopuCATH was evaluated in neutrophil or monocyte/macrophage depletion mice by intraperitoneal injection of anti-Ly6G antibody (clone 1A8, BioXcell) and anti-CSF1R antibody (clone AFS98, BioXCell), respectively. However, I the scavenging efficiency of intraperitoneal injection of anti-Ly6G antibody (clone 1A8, BioXcell) and anti-CSF1R antibody (clone AFS98, BioXCell) should be provided, respectively.

We have provided the scavenging efficiency of intraperitoneal injection of anti-Ly6G antibody (clone 1A8, BioXcell) and anti-CSF1R antibody (clone AFS98, BioXCell) in the revised manuscript as shown in Figure 6—figure supplement 1.